# Unveiling genetic signatures of immune response in immune-related diseases through single-cell eQTL analysis across diverse conditions

Zhenhua Zhang [1,2], Wenchao Li [1,2], Qiuyao Zhan[1,2], Michelle Aillaud[3], Javier Botey-Bataller [1,2,4], Martijn Zoodsma [1,2], Rob ter Horst [5], Leo A. B. Joosten [4,6], Christoph Bock [5,7], Leon N. Schulte [3,8], Cheng-Jian Xu [1,2,4,9], Mihai G. Netea [4,10], Marc Jan Bonder [11,12,13,14] & Yang Li [1,2,4,15,16] ✉

Deciphering the intricate regulatory mechanisms underlying biological processes holds promise for elucidating how genetic variants contribute to immune-related disorders. We map genetic effects on gene expression (expression quantitative trait locus, eQTL) using single-cell transcriptomes of 152 samples from 38 healthy individuals, covering baseline state and lipopolysaccharide challenge either before or after Bacillus Calmette-Guerin vaccination. Interestingly, we uncover a monocyte eQTL linked to the *LCP1*, shedding light on inter-individual variations in trained immunity. Furthermore, we elucidate genetic and epigenetic regulatory networks of *CD55* and *SLFN5*. Of note, our results support the pivotal roles of *SLFN5* in COVID-19 pathogenesis by incorporating disease-associated loci, chromatin accessibility, and transcription factor binding affinities, aligning with the established functions of *SLFN5* in restricting virus replication during viral infection. Our study provides a paradigm to decipher genetic underpinnings of complex traits by integrating single-cell eQTLs with multi-omics data from patients and public databases.

Genome-wide association studies (GWAS) identified numerous genetic variants linked to diseases. However, a large proportion of the loci reside in non-coding regions (NCRs) and consequently require efforts them in disease pathogenesis. These NCRs are usually characterized by regulatory potential to alter gene expression, and abnormal regulations of gene expression can ultimately result in disease[1–3]. Identifying expression quantitative trait locus (eQTL) is promising to dissect gene regulation mechanisms at NCRs harboring GWAS variants. eQTLs are identified by associating gene expression (eQTL-associated gene or eGene) to genetic variants (eQTL-associated variant or eVariant). Large-scale studies presented a comprehensive landscape of associations between variants and gene expression by bulk RNA sequencing providing insights into genetic architectures of gene regulation[3–6]. Recently, single-cell RNA sequencing (scRNA-seq) technologies have opened new avenues for investigating cell-type-specific risk variants and their effects on downstream gene expression, promoting our understanding of the heterogeneity of transcriptomic regulations[7–10].

These advancements underscore the significance of integrating single-cell eQTL (sc-eQTL) with GWAS to unravel genetic heterogeneities underlying diseases. For instance, a population-based study demonstrated that genetic loci contribute to autoimmune diseases by modulating gene expression cell-type specifically[11]. Furthermore, gene expression is context-specific and influenced by dynamic regulations triggered by internal or external stimuli, including inter-/intra-cellular

signals and microbial ligands, inducing spatiotemporal heterogeneities. In addition, the immunological pathogenicity of eQTLs is cell-state dependent, which was demonstrated by profiling dynamic eQTLs during the activation and differentiation of immune cells, such as helper and memory T cells[12,13]. Finally, international collaborative efforts have been established to gain insights into context-dependent gene regulations by interpreting disease-related genetic variants at single-cell resolution[14].

The genetic basis of complex immune traits is poorly known. One important characteristic of innate immune cells, such as monocytes, the capacity to build antigen-agnostic immune memory or "trained immunity (TI)"[15], is characterized by an enhanced response upon homologous or heterologous restimulations. Previous studies suggested that both in vitro stimulation[16] and in vivo BCG vaccination[17] can induce TI that is heterogeneous at the single-cell transcriptome level. However, the roles of genetic variants on gene expression alteration in the TI context are not well-understood.

Previous studies focused primarily on identifying eQTLs at baseline in unperturbed individuals, potentially missing critical insights into the regulatory mechanisms underlying immune response and infection. To comprehend the potential impact of environmental factors - encompassing both internal factors (e.g., cell types) and external factors (e.g., stimuli) - on the complex interplay between genetic variation and gene expression, it is imperative to conduct comprehensive investigations of eQTLs within distinct cell types, while accounting for the type of stimulation or infection.

In this study, we adopted an integrative approach. Initially, we dissected the inter-individual variation in gene expression and regulation, intricately shaped by genetics and environmental factors, in a healthy cohort. These environmental factors impact both in vivo responses induced by vaccination and ex vivo stimulations by stimuli, all at a single-cell resolution. We performed TI eQTL analysis in monocytes and demonstrated that an *LCP1* eQTL contributes to the inter-individual variation in TI that is characterized by fold change (FC) of cytokine response upon re-stimulation. Subsequently, we performed sc-eQTL analysis across conditions at the single-cell level and integrated identified sc-eQTLs with disease-associated loci, chromatin accessibility profiles, and transcription factor binding affinities (TFBA) data, revealing regulatory programs underlying immune-related diseases within distinct immune cells. Finally, we combined the identified response sc-eQTLs with single-cell multi-omics data from COVID-19 patients and uncovered genetic and epigenetic regulatory mechanisms underlying infectious disease. This integrative analysis included TFBA data from public databases, as well as the identified sc-eQTLs and allele-specific open chromatin effects in the relevant cell types in patients.

In this work, by integrating sc-eQTLs with multi-omic data, we systematically unravel regulatory mechanisms resulting from interplays between genetic variants, environmental factors, and gene expression. This elucidates their combined roles in disease susceptibility and inter-individual variations of immune responses. Our study also demonstrates applications of sc-eQTLs identified in healthy donors across diverse conditions, not only in prioritizing candidate genes within GWAS loci but also in unraveling regulatory networks underlying immune response and diseases. To facilitate wider dissemination of our findings, the identified single-cell TI-dependent, consistent and response eQTLs are compiled and are freely accessible at https://lab-li.ciim-hannover.de.

## Results

### Overview of cohort and single-cell RNA-seq data

Totally, 325 healthy volunteers were included in the 300BCG vaccination cohort[18]. In our study, a subset of the cohort consisting of 20 females and 18 males were selected for single-cell transcriptomic profiling[17,18] (Supplementary Fig. 1a, b). Of note, the genotype PCA plots including donors of current study and 1000 genome (phase 3) donors suggested a fair homogeneity of the study cohort (Supplementary Fig. 1c). Peripheral blood mononuclear cells (PBMCs) were isolated from whole blood both before (T0) and three months after vaccination (T3m) as described before[17,19]. Subsequently, the isolated PBMCs were cultured in culture medium with or without lipopolysaccharide (LPS) for four hours ("Methods"). In total, 152 samples across four conditions (RPMI and LPS treated, either before or after vaccination) were profiled using the 10x Genomics Chromium platform (Fig. 1a). These analyses resulted in transcriptional profiles of immune cells across different contexts (in vivo and in vitro stimulations).

The scRNA-seq resulted in transcriptomic profiles of 186,341 cells, which were further classified into seven major cell types: monocytes, CD4+ T cells, CD8+ T cells, NK cells, B cells, dendritic cells, and platelets (Fig. 1b, c, Supplementary Fig. 1e and 1e, Methods). We partitioned the transcriptional variance using a random-effect model and found that cell lineage was the primary determinant, followed by condition and inter-individual variation (Fig. 1e). This underscores the importance of studying gene expression at cellular levels. Notably, no significant differences in cell proportions were observed across the conditions in each cell lineage (Fig. 1e and Supplementary Fig. 1f). Among these identified cell types, we selected five most abundant (≥10%) cell types (monocytes, CD4+ T, CD8+ T, NK, and B cells) for eQTL analyses.

### TI eQTL of LCP1 explains inter-individual variation of TI cytokine response

TI is defined as long-term memories in innate immune cells, leading to an enhanced immune response, such as cytokine production, evoked by the subsequent homologous or heterologous restimulation[20]. Our previous work has shown that the effects of TI are partly genetically regulated[21], while upon restimulation, noticeable alteration of the TI effect was observed at the single-cell transcriptome level[17]. However, how genes contribute to the interindividual variation of TI cytokine responses is largely unknown. To address this, we initially identified 120 TI-response genes and mapped TI eQTL by associating genetic variants with expression alterations, cytokine production fold-change (FC) between T3m-LPS and T0-LPS, of these TI-response genes (Fig. 2a and Supplementary Fig. 2a, Methods).

We compared the identified TI eQTL with TI cQTL effects obtained from the same cohort (Supplementary Data 1). Interestingly, the top locus *LCP1*-rs2806897 (p-value = 3.09e-5, Fig. 2b) out of 118 significant TI eQTL in *cis* (adjusted-p-value < 0.05) was also associated with a TI cytokine QTL (TI cQTL, p-value = 0.04, Fig. 2a, c, and Supplementary Fig. 2b). Importantly, the rs2806897-T allele corresponds to both increased *LCP1* expression FC and TI cytokine (IL6) response FC. The monocyte expression of *IL6R*, which encodes the receptor of IL6, is suggestively associated with the same SNP (rs2806897, P-value = 0.023, Fig. 2d). Collectively, these observations suggest a potential role of *LCP1* in regulating BCG-induced TI-associated cytokine IL6 response.

The *LCP1* gene encodes plastin-2, which plays roles in the activation of innate and adaptive immune cells. Previous studies identified pathways involved in TI, including mTOR, Akt, HIF1A, IL1-family, and MAPK pathways[22]. We calculated transcriptome scores of each of these pathways (Methods) in monocytes and assessed their correlation with alterations of *LCP1* expression upon TI. Interestingly, all of these TI-related pathways exhibited a consistent positive correlation pattern with the alteration of *LCP1* expression (mTOR: cor = 0.35, p-value = 0.031; Akt: cor = 0.38, p-value = 0.02; HIF1A: cor = 0.35, p-value = 0.029; IL1-family: cor = 0.39, p-value = 0.015; MAPK: cor = 0.35, p-value = 0.030, Fig. 2e). This result aligns with previous findings that active *LCP1* interacts with mTOR pathway, further contributing to the mTORC2 activity[23]. Collectively, these data suggest that the TI eQTL of *LCP1* contributes to inter-individual variations in the TI effect induced by BCG, which is summarized in Fig. 2f.

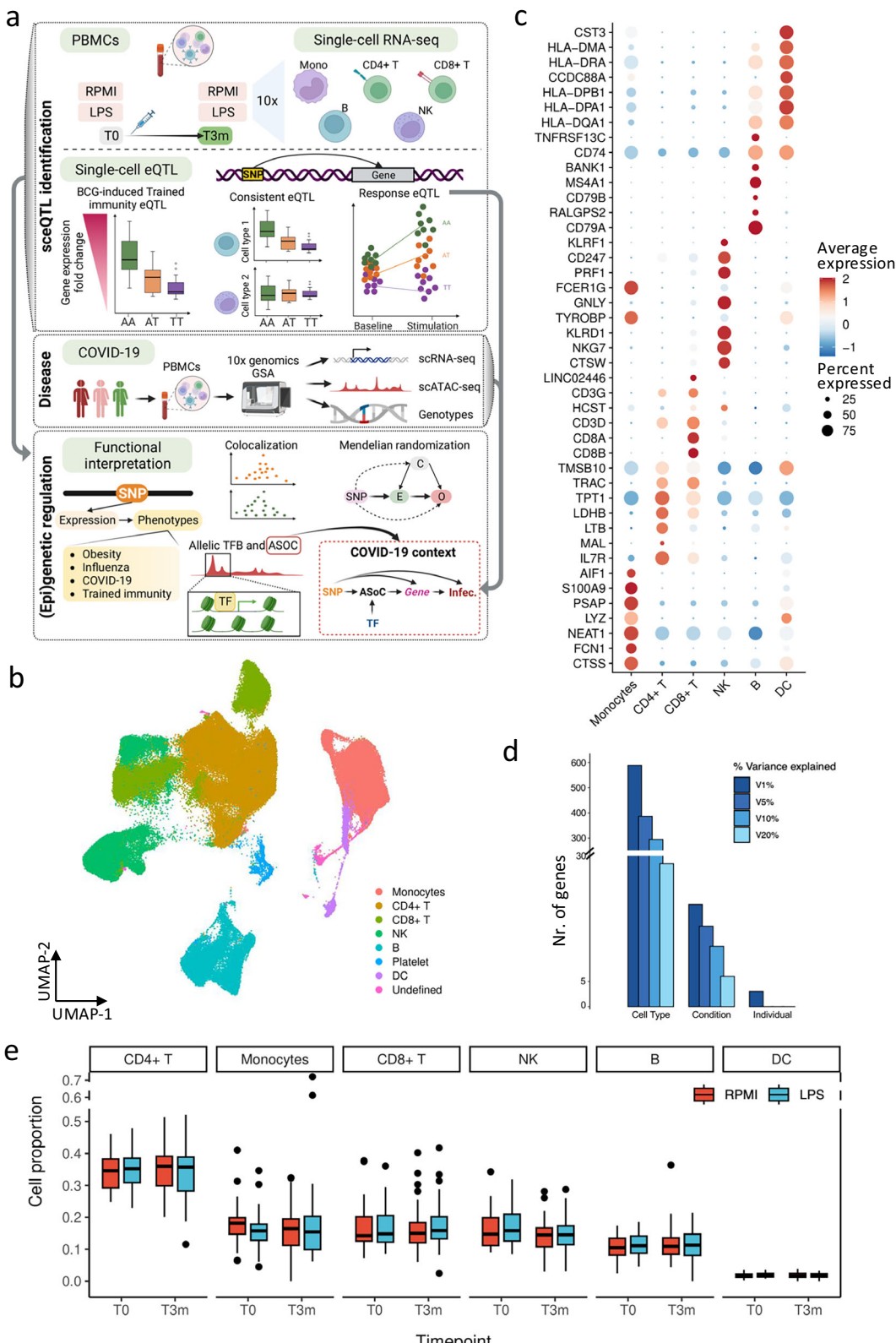

**Fig. 1 | Data set overview. a** Overview of the current study (Created in BioRender. Zhan, Q. (2025) https://BioRender.com/c28r787). The single-cell RNA-seq data were from Li et al.[17] and Zhang et al.[25]. **b** Major cell types identified from the single-cell RNA-seq results. **c** Marker genes used to identify the main cell types in current studies. **d** Variance explained by cell types, conditions, and individuals. The color represents variance explained at 1, 5, 10, and 20% of the gene expression. **e** Cell lineage proportions of each major cell type per individual (*n* = 152 samples). In box pots, whiskers expand to minima and maxima, boxes indicate 25th and 75th percentiles, centers represent medians.

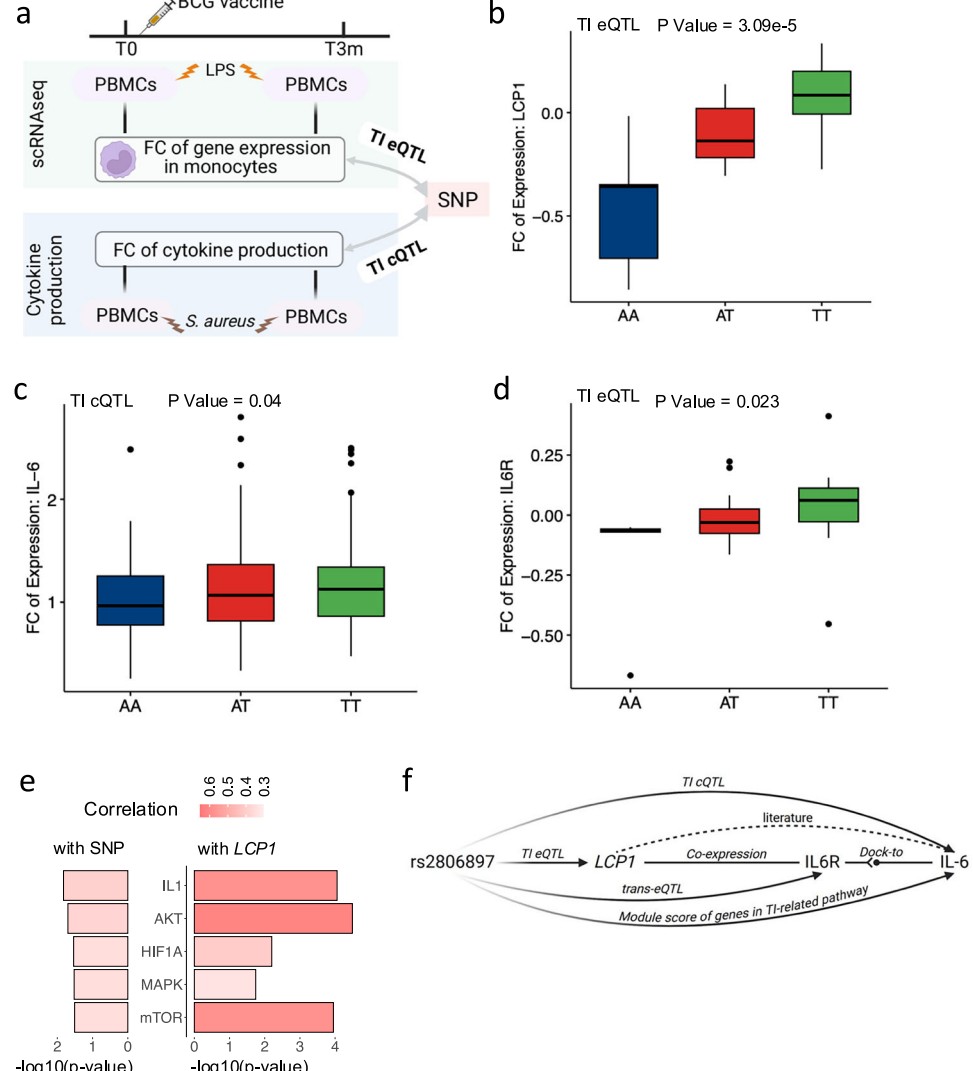

**Fig. 2 | Trained Immunity eQTL (TI eQTL) in monocytes. a** Schematic plot of the TI eQTL of this study. The single-cell RNA-seq data were from Li et al.[17]. **b** Boxplot of leading TI eQTL, *LCP1*-rs2806897 (*p*-value = 3.09e-5, beta = 0.33, 95% confidential interval (CI) = [0.17, 0.49], *n* = 152 samples, LMM adjusting age and sex), in monocytes. The *y*-axis was the fold change (FC) of *LCP1* expression level between after and before being trained. **c** Boxplot of TI cQTL (IL6-rs2806897, *p*-value = 0.04, beta = 0.10, 95%-CI = [0.06, 0.14], DF = 274, *n* = 278 donors, linear regression adjusting age and sex). The *y*-axis was the FC of IL-6 cytokine expression level between after and before being trained. **d** Boxplot of eQTL (*IL6R*-rs2806897, *p*-value = 0.023, beta = 0.20, 95%-CI = [0.05, 0.35], *n* = 152 samples, LMM adjusting age and sex). The *y*-axis was the FC of *IL6R* between after and before being trained. In box-pots (**b**–**d**), whiskers expand to minima and maxima, boxes indicate 25th and 75th percentiles, centers represent medians. **e** Correlation between rs2806897 (left)/LCP1 (right) and TI-related pathways. The *x*-axis represented -log10(p-value) while the color represents correlation strength. Correlations are estimated by the Pearson correlation coefficient. **f** Mechanism of leading TI eQTL (*LCP1*-rs2806897), in monocytes (Created in BioRender. Zhan, Q. (2025) https://BioRender.com/c28r787). In concrete, rs2806897 is associated with *LCP1* gene expression in the TI context, and both SNP and eGene were correlated with TI related pathways. *LCP1* was also correlated with *IL6R* (*p*-value = 2.39e-5, Pearson correlation coefficient = 0.63), which was the receptor of IL-6, while rs2806897 was demonstrated to be associated with IL-6 from the cytokine-QTL analysis.

Furthermore, a previous proteomics study revealed the role of LCP1 in COVID-19 patients[24]. Reanalysis of single cell transcriptome data from three independent COVID-19 cohorts[9,25] revealed that the *LCP1* gene exhibited a significant elevated expression in monocytes in individuals infected by SARS-CoV-2 compared to healthy or convalescent donors (Supplementary Data 2). This result implies that the BCG-induced TI effect of *LCP1* could play a potential role in COVID-19 pathogenesis.

### Consistent eQTLs colocalize with GWAS signals of immune-related diseases

Next, we proceeded to detect consistent eQTLs and response eQTLs. Concretely, we used linear mixed models to estimate associations between genetic variants and gene expression by incorporating

152 samples from four conditions ("Method"). We detected consistent eQTLs by evaluating genetic effects shared across conditions, whereas response eQTLs were identified by testing the interaction between genotypes and conditions (Fig. 1a).

In total, more than 600 consistent eGenes (ranging between 651 to 880) were identified in each cell lineage (Fig. 3a). The varying number of eGenes across cell lineages might be partially due to differences in both proportions of the cell lineage (Supplementary Fig. 1f) and the number of expressed genes in each cell lineage (Supplementary Fig. 1g). These consistent eQTLs were successfully replicated in a publicly available bulk RNA-seq eQTL database (eQTLGen[5]), with concordance rate ranging from 81% to 87% (*p*-value < 0.05 in the replication dataset, Fig. 3b and Supplementary Fig. 3a, b). We also validated our eQTL findings in eQTL datasets from purified specific

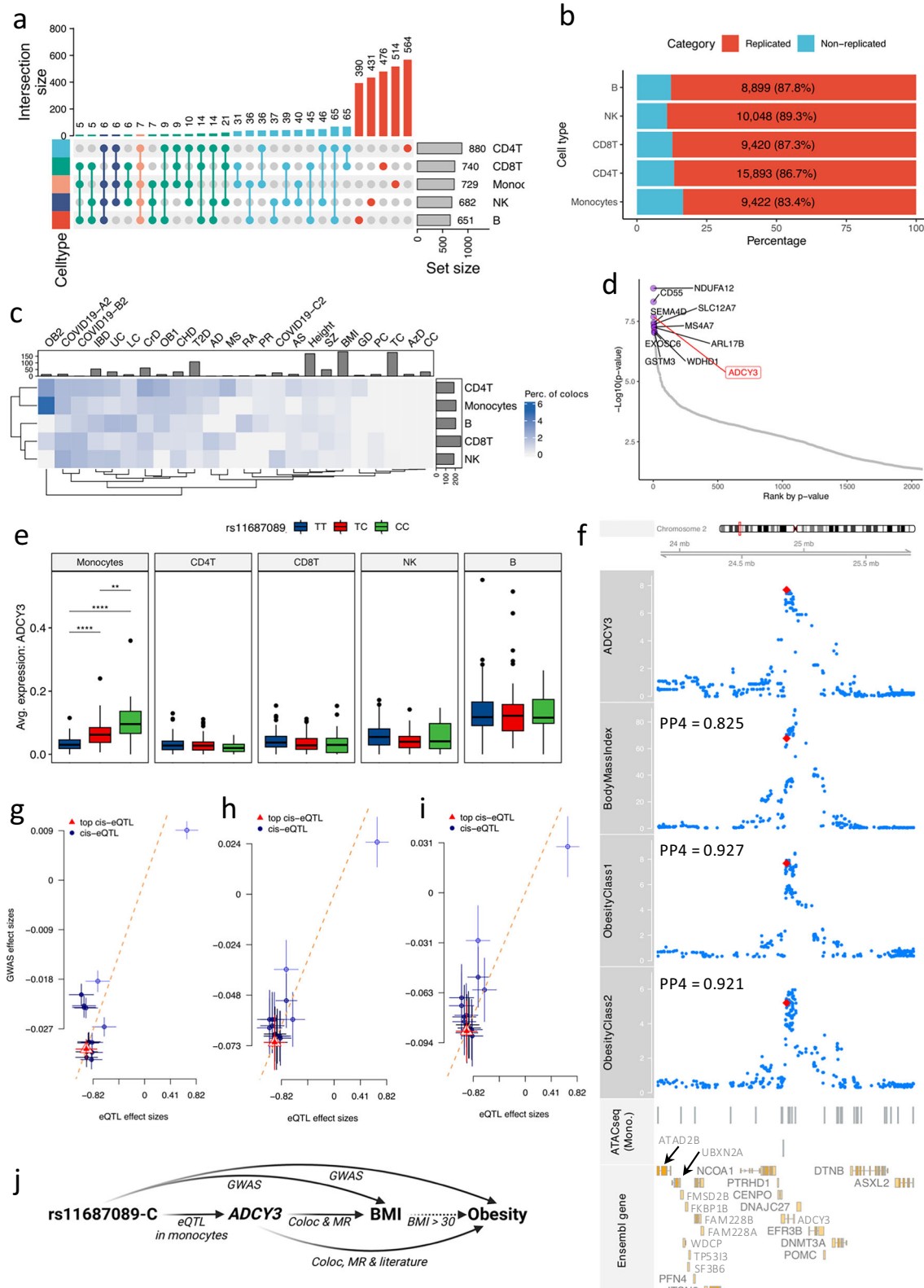

immune cells (DICE database[26]), which resulted in more than 90% validation rates (Supplementary Fig. 3c). Furthermore, a substantial proportion of eGenes are specific to individual cell types, indicating distinct transcriptional profiles in different immune cell lineages (Fig. 3a), which aligns with a previous study[1]. In addition, a further multivariate adaptive shrinkage (MASH[27]) analysis revealed that consistent eQTLs from similar cell lineages exhibit comparatively greater similarity in the genetic effects on gene expression (Supplementary Fig. 3d, e).

Subsequently, in order to assess potential roles of consistent eQTLs in immune traits and diseases, we colocalized sc-eQTLs (Fig. 3c) with publicly available GWAS results (Supplementary Fig. 3f). The results indicate the presence of shared causal variants across multiple complex traits and diseases (Supplementary Fig. 3g). These findings

**Fig. 3 | Monocyte-specific eQTL pinpointed the role of ADCY3 in obesity.**
**a** Number of identified eGenes significantly associated with at least one locus.
**b** Number of eVariants validated in eQTLGen. **c** Colocalization analysis between
consistent eQTL and 24 published GWAS summary statistics. The top bar plot
represents the number of independent loci. The right bar plot represents the fre-
quency of eQTL colocalized with at least one GWAS result. Color in the figure
represents the ratio of colocalized loci out of independent loci per GWAS result.
**d** Top 10 consistent eQTL in monocytes ranked by p-value. Y-axis is -log10-trans-
formed p-value, and x-axis represents p-values' ranks. The top 10 eQTL were
colored (purple) and labeled. ADCY3 gene was highlighted. **e** eQTL effect tagged by
rs11687089 per cell-type (p-value = 2.06e-08, beta = − 0.93, CI = [− 1.27, − 0.57],
n = 152 samples, LMM adjusting age and sex, for monocytes). The Y-axis is the
average expression of ADCY3 per donor. Asterisks indicate significant levels (**
< 0.01, *** < 0.001, **** < 0.0001). In box-pots, whiskers expand to minima and
maxima, boxes indicate 25th and 75th percentiles, centers represent medians.

**f** ADCY3 eQTL in monocytes and body mass index (BMI), obesity class I, and obesity
class II. Red and blue dots represent the top and other SNPs of ADCY3 eQTL. The y-
axis represents -log10-transformed p-values. The X-axis is genomic coordination
(GRCh38). Segments ("ATACseq (Mono.)") represent open chromatin peaks in
isolated monocytes from the cohort superseding our cohort. "Ensembl gene" panel
displays genes in the locus. The PP4 is estimated by Bayesian colocalization analysis
using Bayes Factors. **g–i** Summary data-based Mendelian Randomization analyses
between ADCY3 eQTL and BMI (**g**), obesity class I (**h**) obesity class II (**i**). The y-axis
shows eQTL effect sizes. The X-axis represents GWAS effect sizes. The dashed red
line indicates linear fit of Wald-ratio-test. The top cis-eQTL SNP and rest ones are in
red and blue, respectively. Estimated p-values of SMR are 8.78e-08 (BMI), 9.14e-05
(obesity class I), and 3.90e-04 (obesity class II) using n = 16 SNPs. Vertical and
horizontal error bars are GWAS and eQTL effect sizes (means +/− SD, 95%-CI).
**j** Schematic plot depicting potential roles of ADCY3 eQTL in BMI and obesity
(Created in BioRender. Zhan, Q. (2025) https://BioRender.com/c28r787).

highlight the potential role of genetic variants on complex traits in a
cell-type-specific manner.

Moreover, similar patterns were also observed for identified
response eQTLs (Supplementary Fig. 4a–c). Interestingly, 10.56% –
29.17% of differentially expressed genes (adjusted-p-value < 0.05) are
response eGenes (FDR < 0.05) between the corresponding condition
and baseline (Supplementary Fig. 4d). We also observed correlations
between number of identified responses eQTLs and abundance of a
cell type, which is in line with previous study[28]. Interestingly, we
acknowledge that the number of eGenes in the B cell subset is smallest
in consistent eQTLs (Fig. 3a) but largest in response eQTLs (Supple-
mentary Fig. 4a). This is in line with established findings[29,30], LPS sti-
mulation triggers B cell proliferation and differentiation into antibody-
secreting cells, which likely accounts for the observed expansion of
eGenes in response eQTLs.

The comparison between response eQTLs and consistent ones
suggested specificity in the consistent eQTLs relative to response
eQTLs (Supplementary Fig. 4e). The response eQTLs identified under
different stimulation conditions (LPS response eQTL before BCG or
after BCG) exhibited notable similarity. Besides, we observed shared
eQTL signals across cell types and conditions (Supplementary Fig. 4b).
The shared eQTL signals across cell types and conditions suggest a
conserved genetic architecture that governs gene expression regula-
tion. This implies a broader regulatory influence of these loci across
diverse biological contexts, highlighting their potential roles in coor-
dinating molecular pathways and cellular functions across different
cell types and states.

Finally, the effect sizes and directions of TI eQTLs show correlated
patterns with those of response eQTLs but show less similarity with the
effect and directions of consistent eQTL in monocytes (Supplementary
Fig. 4f). Given that TI assesses the differential response between LPS
stimulation with BCG training (T3m-LPS) and without BCG training
(T0-LPS), it is noteworthy that genetic effect on TI (the difference
between T3m-LPS and T0-LPS) align with the genetic effect on the LPS
response (either T3m-LPS or T0-LPS) for these identified TI genes.

### Shared genetic regulation between ADCY3 expression in monocytes and obesity

The observation of a high proportion of loci shared between monocyte
eQTLs and obesity (Fig. 3c) instigated an exploration into the plausible
contributions of eQTLs to the disease. As indicated in Fig. 3d and
Supplementary Fig. 5a–c, and Supplementary Fig. 6a–d, ADCY3 is an
eGene that are significantly associated with rs11687089. This eVariant
has been previously reported to be correlated to both body mass index
(BMI) and obesity[31,32] (p-value = 1.18e-35 and 2.10e-08, respectively). Of
note, our eQTL analysis revealed that rs11687089-C corresponds to
increased ADCY3 expression predominantly in monocytes (Fig. 3e). A
colocalization analysis between the ADCY3 eQTL in monocytes and
publicly available GWAS summary statistics including BMI, obesity

class I (OB1), and obesity class II (OB2)[32] revealed that the identified
locus is associated with both ADCY3 expression in monocytes and
obesity-related traits with high probability (PP4$_{eQTL vs BMI}$ = 0.825,
PP4$_{eQTL vs OB1}$ = 0.927, and PP4$_{eQTL vs OB2}$ = 0.921, Fig. 3f). A further
summary statistic-based Mendelian Randomization (MR) analysis
using the SMR method[33] showed consistently significant mediation
effects on BMI/obesity through the expression of ADCY3 in monocytes
(p-value$_{eQTL vs BMI}$ = 8.78e-08, p-value$_{eQTL vs OB1}$ = 9.14e-05, and p-
value$_{eQTL vs OB2}$ = 3.90e-04, Fig. 3g–i and Supplementary Fig. 6e–g,
"Methods"). As summarized in Fig. 3j, these observations suggest the
potential involvement of ADCY3 in increasing BMI and the develop-
ment of obesity, expanding previous studies[34,35].

### Allelic transcription factor (TF) binding affinities contribute to CD55 eQTL effect in monocytes

One of the interesting observations was the significant association
between rs2564978-C and CD55 expression in monocytes but with
opposite eQTL effect direction in other cell types (Fig. 4a and Sup-
plementary Fig. 7a, b). This monocyte eQTL matches previous bulk-
eQTL studies (GTEx and DICE database[4,26], Supplementary Figs. 8a, b,
9a) but sheds light on its cell-type specificity. In addition, using
chromatin accessibility data of PBMCs (299 samples) and from
monocytes (27 samples) from the 300BCG cohort[21,36], we found an
open chromatin region (OCR) harboring eVariant associated with
CD55 gene expression (Fig. 4b and Supplementary Fig. 8b) in PBMCs
and monocytes. Furthermore, the potential regulatory programs
between the variant and CD55 was also observed using gene-enhan-
cer/promoter interactions from the GeneHancer database[37] (Sup-
plementary Fig. 8b).

Genetic variants in regulatory regions (promoters and enhancers)
can modulate gene expression by altering binding affinities of tran-
scriptional factors, which consequently disrupts the expression bal-
ances between alleles and offers means to interpret eQTL effects[38].
Therefore, we proceeded to test if the identified eVariant could per-
turb CD55 expression by allelically altering TF binding affinities. Initially,
we obtained the potential allele-specific TF binding affinities of the
eVariant in immune-related cells from the ADASTRA database[38]. We
observed significant higher allelic binding affinities to allele rs2564978-
C in classical monocytes, neutrophils from peripheral blood, and dif-
fuse large B-cell lymphoma (Fig. 4c). By using the same database, 10
TFs were prioritized by their binding capacity to the identified OCR
and overlapped with the identified eVariant (Fig. 4d, Methods). Of
these TFs, SPI1, STAT1, ESR1, EP300, CEBPA, and CTCF display sig-
nificantly higher binding affinities to the allele rs2564978-C (FDR <
0.05), which could result in an increased expression of CD55 com-
pared to those in alternative allele carriers. Moreover, the reference
allele has a significant capacity to interrupt the binding motifs of SPI1,
CEBPA, and CTCF (Fig. 4d), suggesting their potential regulatory roles
in modulating the CD55 expression.

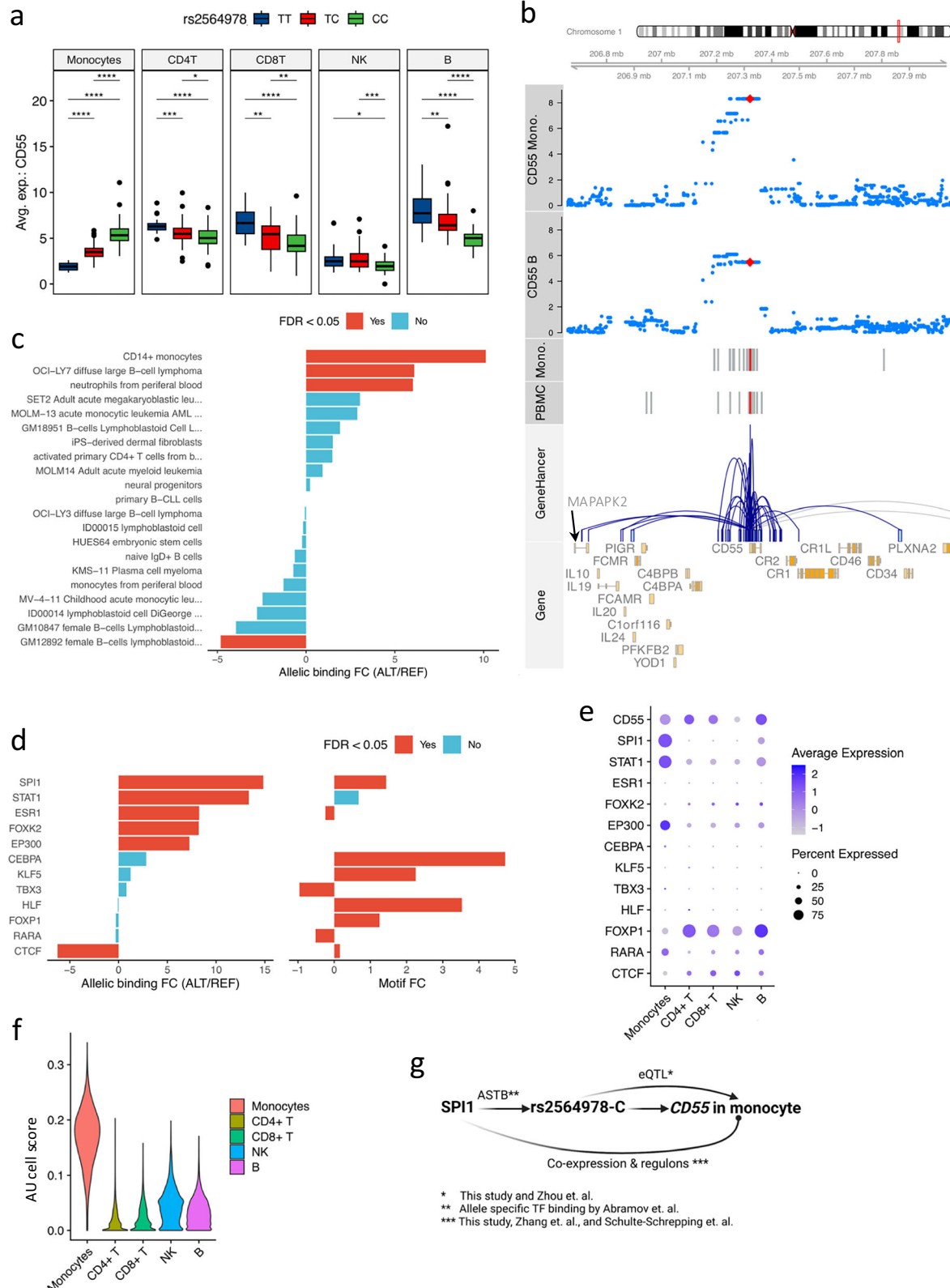

Furthermore, the expression profiles of these TFs across cell types suggested SPI1, STAT1, and EP300 are potential key regulators of *CD55* in monocytes (Fig. 4e). Among the prioritized TFs, we assessed their effects on *CD55* expression using regulon analyses[39]. TF regulon enrichment, assessed via the AUC (Area Under the Receiver Operating characteristic Curve), revealed that *SPI1* was specifically enriched in monocytes (Fig. 4f), while other TFs did not display cell-type-specific over-representation (Supplementary Fig. 9b). Furthermore, of the evaluated TF and regulons, only SPI1, STAT1, and ESR1 were predicted regulating *CD55*. And the *CD55* was among the first of four quantiles of SPI1's target genes, which was ranked by the regulation potentials of SPI1 (Supplementary Fig. 9c). Notably, a recent study[40] demonstrated that SPI1 modulates *CD55* promoter activities by binding to the rs2564978-C allele, consistent with our results. In addition, the co-

**Fig. 4 | CD55 eQTL explained by allelic transcription factor binding affinities.**
**a** *CD55*-rs2564978 eQTL effect per cell type. The *y*-axis is the average expression of *CD55* per donor. Including monocytes (*p*-value = 4.86e-09, beta = 1.26, 95%-CI = [0.82, 1.71]), CD4 + T (*p*-value = 3.49e-03, beta = − 0.59, 95%-CI = [− 0.99, − 0.19]), CD8 + T (*p*-value = 0.01, beta = − 0.54, 95%-CI = [− 0.97, − 0.11]), NK cells (*p*-value = 3.56e-03, beta = − 0.49, 95%-CI = [− 0.83, − 0.16]), B cells (*p*-value = 3.32e-06, beta = − 1.04, 95%-CI = [− 1.49, − 0.58]). *P*-value < 0.05: *, < 0.01: **, < 0.001: ***, < 0.0001: ****, LMM adjusting age and sex, *n* = 152 samples. Whiskers of box-plots expand to minima and maxima, boxes indicate 25th and 75th percentiles, centers represent medians. **b** Locus plot of *CD55* eQTL effects (monocytes and B cells). The top two tracks are genome-coordination (GRCh38). The next two are p-values of *CD55*-SNP pairs in monocytes and B cells, respectively. The *y*-axis shows -log10-transformed p-values. Red dots are the top SNP of CD55 eQTL in monocytes. Fifth and sixth are OCRs (monocytes and PBMC). Red bars are OCRs overlapping the top eVariants of *CD55*. Seventh is GeneHancer interactions (UCSC genome browser).

The last track displays gene (ENSEMBL). **c** Allelic TF binding potentials of rs2564978. The *y*-axis displays immune cells, and the *x*-axis shows the log2-transformed FC of allelic binding affinity (alternative allele vs reference allele). Colors indicate significance (adjusted-*p*-value < 0.05, two-sided beta-binomial test). **d** Allelic TF binding affinities of TFs potentially bound eVariant locus. The left panel shows allelic TF-binding of eVariant (adjusted-*p*-value < 0.05, two-sided beta-binomial test). The right panel displays allelic TF brokenness. Colors denote significance. The *X*-axis of both panels are log2-transformed allelic-binding affinity FC and motif-break-score FC (alternative to reference). **e** Expression of prioritized TFs per cell type. Dots colors and sizes represent average expression and percentage of cells expressing the gene, respectively. **f** Violin plot displaying SPI1 regulatory activities per cell type. The *X*-axis represents cell types and *y*-axis represents AUcell scores of SPI1 by SCENIC[39]. **g** Schematic figure showing potential regulation of CD55 expression (Created in BioRender. Zhan, Q. (2025) https://BioRender.com/c28r787).

expression profiles can also be observed in pseudo-bulk PBMC from the public datasets[9,25] (Supplementary Fig. 10).

To validate the identified potential regulation programs, we measured *CD55* expression in monocytes after suppressing SPI1 or STAT1 using the corresponding TF inhibitors (Supplementary Fig. 9d, e). Although no statistically significant effect was found partly due to the limited number of samples in the experiment, we observed a trend towards suppressed *CD55* expressions in monocytes. Of note, a recent publication[40] reveal that the rs2564978-C allele is associated with increased promoter activities upon bound by SPI1, which is in line with our results. Our results collectively provide insights in the regulatory roles of eVariants interfering with TF binding affinities and resulting in eQTL effect in monocytes (Fig. 4g).

**Integration of response eQTLs with multi-omics data of COVID-19 patients unravels genetic and epigenetic regulation of *SLFN5***
External stimuli can significantly alter transcriptional profiles of immune cells, resulting from complex regulatory processes. Therefore, we identified response eQTLs to understand genetic variations associated with transcriptional alterations upon stimulations (Supplementary Fig. 4a). Interestingly, we identified robust response eQTL effects for the *SLFN5* gene in T lymphocytes. Concretely, the rs11080327-A allele was significantly associated with increased *SLFN5* expression in T lymphocytes upon LPS stimulation, both in samples collected before and three months after BCG vaccination (Fig. 5a and Supplementary Fig. 11a, b). Previous studies suggested the potential role of *SLFN5* in controlling viral loads, such as influenza A virus (IAV) and SARS-CoV-2[41,42]. In line with this, our colocalization analysis suggests a potential shared genetic basis between *SLFN5* expression and COVID-19 severities[43] (Fig. 5b, PP4 = 0.398), where the rs11080327-A allele is associated with increased *SLFN5* expression but decreased risk of COVID-19 severity (*p*-value = 9.83e-4). Importantly, this locus harbors a SNP (rs883416) that is suggestively associated with COVID-19 severity (*p*-value = 4.55e-05) and strongly linked to rs11080327 ($R^2$ = 0.965, *p*-value < 0.0001, EUR). In addition, this SNP is located in an open chromatin region (Fig. 5b) based on bulk ATAC-seq data from PBMC of 299 samples in the same cohort (300BCG), suggesting a regulatory role of the identified locus. This potential regulatory program (the interaction between enhancers and promoters) is also observed in the publicly available GeneHancer database[37] (Fig. 5b).

To shed light on the role of *SLFN5* in the COVID-19 context, we estimated the eQTL effects in T lymphocytes using scRNA-seq data from 37 hospitalized and 27 convalescent COVID-19 patients[25] (Fig. 1a). First, we observed significantly elevated expression of *SLFN5* in hospitalized individuals compared to convalescent individuals (adjusted-*p*-value = 3.71e-08, Fig. 5c and Supplementary Fig. 12a), which aligns with the known function of *SLFN5* in controlling viral replication[44]. Furthermore, the rs11080327-A allele was significantly associated with elevated *SLFN5* expression in CD4 + T cells, confirming the response

eQTL effect, which was originally identified in healthy individuals, of *SLFN5* in T cells in a disease context (Fig. 5d). Moreover, this eQTL effect could be replicated using single-cell multiome data from an independent COVID-19 cohort, consisting of 43 patients. As shown in the Supplementary Fig. 12b, the rs11080327-A allele is suggestively associated with higher expression of *SLFN5* in CD4 + T cells lymphocytes (*p*-value = 0.09, "Methods"). To sum up, the *SLFN5*-rs11080327 eQTL effect was (suggestively) replicated in patient data using two independent COVID-19 datasets with *p*-value 4.55e-05 and 0.09, respectively.

To understand how genetic variants regulate *SLFN5* expression, we comprehensively investigated the locus. Specifically, we assessed the chromatin accessibility in CD4 + T cells using single-nuclei ATAC-seq data obtained from the same cohort of COVID-19 patients[25] (Fig. 5e). This analysis revealed an open chromatin peak which harbors the eVariant and observed in CD4 + T cells of hospitalized and convalescent donors. Notably, we observed a significant correlation between this peak and *SLFN5* expression in T cells (Spearman's rho = 0.84, FDR-adjusted-*p*-value < 0.05), suggesting its potential regulatory roles. Subsequently, to determine the allelic effect of this OCR, we examined the allele-specific open chromatin effects in CD4 + T cells from hospitalized COVID-19 patients and convalescent individuals, respectively (Fig. 5f). The significant allelic imbalance of chromatin accessibility suggested that rs11080327-A is responsible for the elevated *SLFN5* expression, a finding consistent with our differential expression analysis (Fig. 5d). In addition, TBX21, IRF4, and STAT1 have high binding affinities to motifs of rs11080327-A allele, indicating that the allele may interfere TF binding (Fig. 5g and Supplementary Fig. 12c–e). These suggest that the eVariant may interfere with the regulatory capacities of these TFs, aligning with the established roles of STAT1 on *SLFN5*[44,45].

To summarize, the integration of response eQTL from health and single-cell multi-omics data of COVID-19 patients unravels the genetic and epigenetic regulations of *SLFN5* expression (Fig. 5h). Moreover, we also observed response eQTLs that are colocalized with complex traits involved in hosts' immune responses, such as type 2 diabetes[46] (Supplementary Fig. 12f, g).

## Discussion
In this study, we systematically estimated genetic contributions to cell-specific gene expressions using single-cell transcriptomes of 152 samples from 38 healthy individuals, covering baseline state and LPS treatment (before and after BCG vaccination). We identified TI eQTLs, consistent eQTLs, and response eQTLs in five major cell lineages from PBMCs. Follow-up integration analyses, involving publicly available GWAS and single-cell multi-omics, unraveled regulatory mechanisms in immune traits and diseases.

First, we identified genetic variants associated with expressions of TIGs in monocytes to elucidate genetic bases of TI. The analyses

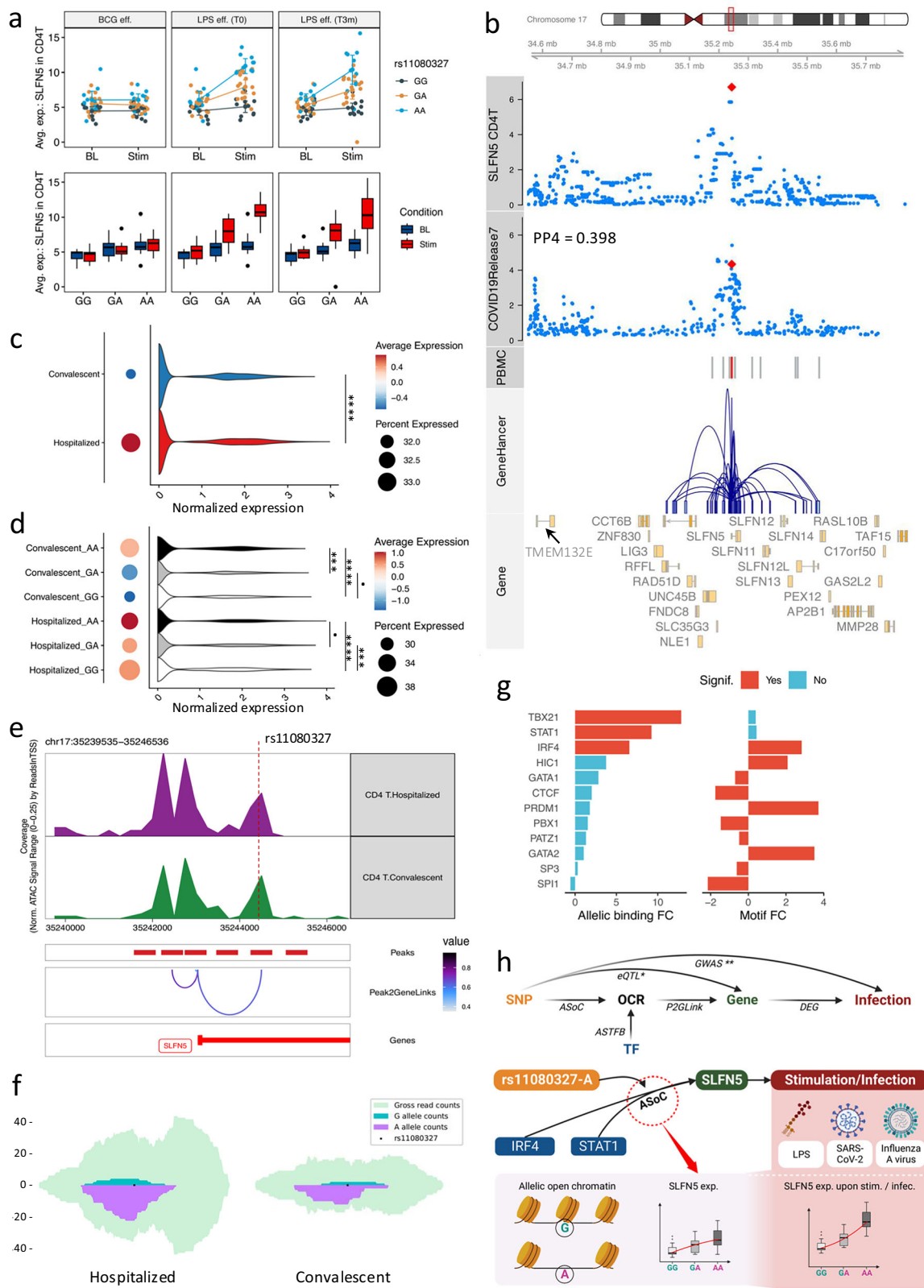

pinpointed *LCP1*-rs2806897 as the top TI eQTL. Of note, rs2806897-A allele carriers exhibited lower *LCP1* expression after training, meanwhile, the TI cQTL revealed that individuals with the A allele displayed poorer induction of trained immunity compared with the individuals harboring T allele[19]. Furthermore, we constructed a regulatory network of *LCP1*, combining genetic, transcriptome, and cytokine production to interpret TI in COVID-19.

Previous studies have suggested a potential interplay among IL-6, IL6R (IL-6 receptor) and LCP-1 in various contexts[47–50]. IL-6 is well known for its role as a warning signal in response to infections, tissue damage, or other stimuli[51]. Upon binding to IL-6R, downstream signaling molecules such as gp130 activate the JAK2/STAT3 pathway and JAK-SHP-2/MAPK pathway, which are central communication nodes for cell functions. Recent studies on osteosarcoma involving the *LCP1*

**Fig. 5 | COVID-19 related SLFN5 gene is regulated via chromatin accessibility in lymphocytes. a** eQTL effects of *SLFN5*-rs11080327. Panels display response eQTL per stimulation: BCG-vaccination, LPS pre-BCG-vaccination, 3-month post-BCG-vaccination. Colors represent genotypes. The *X*-axis shows conditions. The *Y*-axis is *SLFN5* expression. Error bars are mean values +/− SD (95%-CI) and mean expression are connected by line (*n* = 152 samples). Whiskers of boxplots expand to minima and maxima, boxes indicate 25th / 75th percentiles, and centers represent medians. **b** Regulation potentials of *SLFN5*-rs11080327. Track-plot includes chromosome, genome-coordination, eQTL p-values in CD4 + T cells (LMM adjusting age and sex) and COVID-19 GWAS p-values (logistic regression), OCRs (PBMC), gene-enhancer interactions (UCSC genome browser), and gene (ENSEMBL). Red dots (tracks 3, 4) denote rs11080327. Red bars (track 5) denote OCRs overlapping rs11080327. **c** *SLFN5* expression per COVID-19 condition (CD4 + T). Dot colors and sizes indicate average expression and cells (percentage) expressing *SLFN5*, respectively. Violin plots show *SLFN5* normalized-expression (*x*-axis) per cell type (adjusted-*p*-value = 3.71e-08, log2-FC = 0.22, LR-test, adjusted-*p*-value < 0.0001: '\*\*\*\*'). **d** *SLFN5*-rs11080327 eQTL in COVID-19. Dot colors and sizes show average expression and

cells (percentage) expressing *SLFN5* per condition per genotype. Violin plots show *SLFN5* normalized expression (*x*-axis). Convalescent, adjusted-*p*-value $_{GA/AA}$ = 2.26e-04, log2-FC $_{GA/AA}$ = −0.20; adjusted-*p*-value $_{GG/AA}$ = 7.91e-10, log2-FC $_{GG/AA}$ = − 0.24; adjusted-*p*-value $_{GG/GA}$ = 0.28, log2-FC $_{GG/GA}$ = -0.04. Hospitalized: adjusted-*p*-value $_{GA/AA}$ = 0.20, log2-FC $_{GA/AA}$ = − 0.06; adjusted-*p*-value $_{GG/AA}$ = 6.36e-05, log2-FC $_{GG/AA}$ = −0.06; adjusted-*p*-value $_{GG/GA}$ = 1.08e-04, log2-FC $_{GG/GA}$ = 0.01. LR-test (adjusted-*p*-value > 0.05: '-', < 0.001: '\*\*\*', <0.0001: '\*\*\*\*'). **e** OCRs at rs11080327 (COVID-19). The top two tracks display OCRs (CD4 + T). "Peaks" shows OCRs correlated to *SLFN5* expression in "Peak2GeneLinks". The color of connection suggests Pearson's rho correlation. **f** Allelic open chromatin at eVariant in COVID-19 (CD4 + T). Green and purple denote read counts covering G/A alleles. Light blue shows gross read counts. **g** Allelic TF binding affinities (rs11080327). The left panel shows allelic TF-binding affinities. The right panel displays allelic TF brokenness. Colors denote significance (adjusted-*p*-value < 0.05). X-axis are log2-FC of binding affinity and motif-break-score (alternative to reference). **h** Potential regulations of *SLFN5* expression upon stimulation (infection) involving allelic open chromatin (Created in BioRender. Zhan, Q. (2025) https://BioRender.com/c28r787).

gene (also known as actin-bundling protein L-plastin or LPL), have shown that the aberrant expression of *LCP1* gene activates the JAK2/STAT3 signaling pathway, linking IL-6, IL-6R, and LCP1 in the context of tumor progression[47]. Furthermore, LCP1 deficiency in mice is associated with decreased IL-6 production at six hours post-infection with *P. aeruginosa*[48]. This deficiency particularly affects the myeloid compartment, making it more susceptible to infection. Moreover, LCP1 also plays a role in modulating NLRP3-mediated production of IL-1β and IL-18 by enhancing NLRP3 assembly in tissue-resident macrophages[49]. Since IL-6 is a known downstream target of IL-1β, it is consistently increased in serum from patients with NLRP3 inflammasome-mediated conditions[50]. Collectively, these studies highlight potential interplays among LCP1, IL-6 and IL-6R, in line with our observations.

Then, we integrated consistent eQTLs with publicly available datasets related to common diseases to elucidate the roles of eGenes in complex traits. The integration of consistent eQTLs and genetic associations of obesity GWAS validated the role of *ADCY3*, specifically in monocytes. The *ADCY3* gene encodes adenylyl cyclase 3, a membrane-associated enzyme that plays a pivotal role in the cAMP (cyclic adenosine monophosphate) signaling pathway. Previous studies have implicated this enzyme in mediating energy homeostasis, potentially contributing to obesity. For instance, murine models with biallelic or monoallelic *ADCY3* mutations have displayed increased susceptibility to obesity and insulin resistance[52–54]. In addition, biallelic loss-of-function mutations in *ADCY3* have been associated with severe obesity, underscoring its involvement in regulating body weight and the development of obesity[34]. These studies mainly focused on the neuronal cilia and have interpreted the causal role of *ADCY3* through the lens of ciliopathies, offering promising drug targets[55]. Moreover, *ADCY3* is involved in TI through its involvement in cAMP signaling processes within classical monocytes[56]. The elevated numbers of monocytes in peripheral blood is recognized as a marker of severe obesity[57–59], which suggesting high inflammations. These observations hint at diverse roles of *ADCY3* in different tissues, warranting further research efforts to assess its drug target potential.

The intricate control of gene expression relies on the interplay between regulatory DNA elements and promoters of target genes[60]. However, genetic variants residing within enhancers and promoters can disrupt TFs' binding affinities by altering DNA sequences of binding sites[38]. To assess the regulatory potential of the identified eVariants, we examined their capacity to change TF binding affinity in predicted enhancers and promoters, thereby elucidating inter-individual variations of *CD55* expression. Our study identified an eVariant associated with *CD55* expression in monocytes. This eVariant is located within the promoter region of the *CD55* gene and has the potential to disrupt the motif sequences that are putatively bound by

TFs encoded by *SPI1*, *STAT1*, and *ESR1*, where co-expressions between the TF genes and *CD55* were also observed in our data. The *CD55* gene encodes the complement decay-accelerating factor, which inhibits early-stage activation of the complement system[61,62], involving immune processes broadly. Previous studies have also indicated active interplays between the aforementioned TFs, SPI1, STAT1 and ESR1 and *CD55*[63–66]. These findings uncover *CD55* regulatory mechanisms involving multiple transcription factors and illustrate an approach to interpret eQTLs by integrating epigenetic profiles.

Our integrative analyses, involving single-cell response eQTLs, COVID-19 GWAS, and single-cell multi-omics data of COVID-19 patients, has also revealed potential roles of *SLFN5* in CD4 + T cells upon SARS-CoV-2 infection. Previous studies suggested that SLFN5 is potentially involved in COVID-19[67], however, the underlying regulatory program is still unclear. Therefore, we uncovered the epigenetic and genetic regulation mechanisms by integrating multiple evidence, including (1) colocalization by COLOC[68] and Mendelian Randomization by SMR[33]; (2) open chromatin peak harboring the eVariant of SLFN5 measured in the same individuals, i.e., ATAC-seq results; (3) regulatory potentials of identified eVariant from publicly available database such as GenHancer and ADASTRA[38]. First, *SLFN5* is an interferon-stimulated gene and upregulated in response to infection by IAV[41]. Notably, the increased association between the eVariant and *SLFN5* in the IAV-infected cells aligns with our response-eQTL findings, reinforcing *SLFN5's* involvement in response to viral infections[16]. Moreover, our investigation has revealed potential shared association signals between *SLFN5* expression and COVID-19 outcomes[43]. Furthermore, our single-cell ATAC-seq analysis in a COVID-19 cohort[41] suggests that the eVariant modulates *SLFN5* expression by altering chromatin accessibility. Of note, this eVariant may perturb binding affinities of TFs, including TBX21, STAT1, and IRF4. Worth noting, the ChIP-Seq signal[69] has indicated the eVariant represents a regulatory DNA sequence interacting with STAT1, a pivotal mediator of interferon responses. And, the conversion of the heterozygous rs11080327$^{A/G}$ into the homozygous rs11080327$^{G/G}$ state has been shown to restore *SLFN5* gene expression[45]. In light of these findings, our comprehensive analysis supports the involvement of *SLFN5* in viral infections (IAV and SARS-CoV-2) and underscores its genetic regulation mediated by altering chromatin accessibilities.

Previous studies have shown that the LPS-induced T-cell activation is assisted by primed monocytes[70], which is highly depended on the cell-cell communications. In addition, T lymphocytes can also be activated by LPS via TLR4/CD14 complex[71–73], which enhances the ability of T cells to beat bacteria. To explore whether the eQTL effect of rs11080327 on *SLFN5* is also present in monocytes under stimulation, we analyzed our dataset and observed this eQTL effect in monocytes. These findings suggest potential intercellular interactions acting in

trans between monocytes and T cells in response to LPS stimulation. However, the precise direction and mechanism of these interactions will require experimental validation in future studies. Of importance, the above results emphasized the role of identifying response eQTLs in specific contexts to improve our knowledge on genetic architectures that modulate immune responses, which is also proposed by other studies[10,11,28,74].

The findings and workflow in this study provided examples to validate and prioritize inheritable units (i.e., genes) in complex traits by incorporating multi-omics data at cellular resolution. However, limited by the sample size, some of our findings are underpowered but still indicative, which could be improved by large-scale collaborative efforts. In the study, we exploited single-cell transcriptional profiling techniques, which enable researchers to zoom into individual cells. However, these methods are restricted by the nature of gene expression, i.e., only a small set of genes are expressed to maintain cell life and conduct their designated functions. Moreover, our sc-eQTL analyses were focused on major immune cell lineages in PBMCs, incorporating limited environmental factors, however more efforts are needed to elucidate the genetic architectures of gene expression in less prevalent cell types such as DCs. Although we reconstructed a potential regulatory network involving TI in monocytes from COVID-19 patients, we did not detect genome-wide significant signals, possibly due to limited sample sizes. In addition, further investigation is needed to reveal the genetic regulatory relationship in other innate immune cells, such as NK cells and dendritic cells. And, CD8 + T cells may interact with monocytes during training[75], therefore, the interpretation of this finding from the genetics perspective is also worth exploring.

We recognize that exploring the TI effect in less abundant innate cell types, such as NK cells, will require a larger sample size. Future studies could leverage single-cell analysis combined with cell sorting or enrichment strategies to examine TI effects in neutrophils and other low-abundance innate immune cells[11–13]. Applying the integration approach demonstrated in this study to such high-resolution datasets would enable researchers to gain a more comprehensive understanding of context-specific eQTLs and their associations with disease.

In this study, we reused scRNA-seq data from our previous work[17], including cell type annotations and donor assignments, to explore genetic regulation of gene expression via sc-eQTL analysis. Shifting focus from immune transcriptional responses to TI, we investigated genetic and epigenetic influences under both in vivo and in vitro conditions. Instead of focusing on the monocyte heterogeneity in TI, our current work offers a framework to uncover genetic mechanisms of complex traits through integrated sc-eQTL and multi-omics analyses.

In summary, we identified sc-eQTLs in PBMC and interpreted their potential roles in complex traits such as common diseases, immune responses, and infectious diseases, using four examples of eQTLs. These eQTL are valuable resources to understand the genetic drivers of gene expression upon stimulation at individual cell type levels. The integration results provide an example workflow to interpret complex trait loci.

## Methods

### Study approval
The 300BCG (NL58553.091.16) study was approved by the Arnhem-Nijmegen Medical Ethical Committee. Written informed consent was obtained before any research procedure was initiated. The study was performed in accordance with the declaration of Helsinki.

### Cohort information
To study the immunological effects of BCG vaccination, 325 healthy (44% male and 56% female) adult volunteers of Western European ancestry were included in the 300BCG cohort between April 2017 and June 2018 at the Radboud University Medical Center. Healthy volunteers were recruited using local advertisements and flyers in Nijmegen (the Netherlands) and were compensated for participation. After written informed consent was obtained, EDTA blood was collected, followed by administration of a standard dose of 0.1 mL BCG (BCG-Bulgaria, InterVax) intradermally in the left upper arm by a medical doctor. Vaccination of study participants was organized in batches of 6–16 subjects per day. Three months after BCG vaccination, additional blood samples were collected. The exclusion criteria comprised the use of systemic medication, except for oral contraceptives or acetaminophen, and antibiotics within three months before inclusion, previous BCG vaccination, history of tuberculosis, any febrile illness four weeks before and during participation, any vaccination within three months before participation, and a medical history of immunodeficiency. The fold change (FC) of IL-6 between T3m and T0 was used as a measure for BCG-induced TI.

### PBMC isolation and Single-cell RNA-seq data generation
The isolation of PBMCs and simulations were done as described by Li et al.[17]. In brief, PBMCs were collected before and 3 months post 0.1 mL BCG vaccination (BCG vaccine strain Bulgaria; Intervax), followed by ex vivo stimulation with RPMI medium (control) or 10 ng/mL LPS. Then, 152 samples were randomly pooled into 32 libraries with equal cell numbers from 3/5 samples in each pool. We used the 10X Genomics Chromium platform to generate single-cell gene expression libraries by using the Chromium Next GEM Single Cell 3′ Library & Gel Bead Kit v3.1 and Chromium Next GEM Chip G Single Cell Kit (10x Genomics) according to the manufacturer's protocol. NovaSeq 6000 S4 was used for sequencing with 28 bp R1 and 90 bp R2 run settings. In order to minimize the batch effect, 32 pools were split into 3 batches randomly.

### Genotyping, imputation, and quality control
Genotyping, imputation, and corresponding quality control were described by Kong et al.[19] and Li et al.[17]. However, we subsetted genetic variants for donors involved in the current study, and quality controls for the variants were performed as the following. In brief, we obtained genetic variants of high quality by excluding genetic variants that 1) missing rsID, and 2) with less than three homozygous or heterozygous genotypes using BCFtools (version 1.12)[76].

### Single-cell RNA-seq pre-processing
Demultiplexing of samples and cell type annotation were performed as described in the previous study[17,19]. In detail, we converted BCL files to FASTQ files using bcl2fastq Conversion Software offered by Illumina. Sequence data was aligned to GRCh38/b38 using 10X Genomics STAR in the CellRanger pipeline (v3.1.0)[77,78]. Souporcell (v2.0)[79] was used to remove doublets and to assign singlets to their individual of origin. The Seurat (v4.1.1)[80] package in R was used to integrate single-cell RNA-seq from 3 batches. Cells were excluded if the mitochondrial gene percentage > 25%, and the number of detected genes < 100 or > 5000. NormalizedData() and FindVariableFeatures() functions with default parameters were used to normalize data and identify variable features. Then, ScalaData() was used to scale the data, and then RunPCA() for dimension reduction. Finally, FindIntegrationAnchors() was used to integrate all batches. In addition to manually selecting known markers, we also used SingleR[81] for automatic cell type annotation. In total, 6 main cell types (CD4 + T cells, CD8 + T cells, monocytes, NK cells, B cells, and DCs) were identified and annotated for downstream analysis.

### Variance component analysis
We adopted the strategy proposed by ref. 82. For each gene, variance component analysis was performed by fitting gene expression using a random effect model in LIMIX. We randomly selected 5000 cells to

reduce the computation burden. The cell type, condition, and individuals were selected as random effects in each model.

## Gene expression profile

The single-cell level gene abundance profiles were represented using the pseudo-bulk method, i.e., the mean expression per cell type per donor, which was calculated by the AverageExpression() function from the Seurat package. To obtain high-quality eQTL, only genes with non-zero expression in at least 30% of donors were retained for the downstream analysis, which resulted in 9689 genes.

## Identifying cell-type and response eQTL

Subsequently, to identify genes whose expressions are associated with genetic variants, or the so-called eGene, we evaluated the correlation between the expression of selected genes and SNPs that are located in a 1 Mb base-pair window around the transcription start sites (MAF > 0.05), using a linear mixed model (LMM) implemented in the limix_qtl package[83]. In the model, age, sex, stimulation, and BCG vaccination were included as co-variables. To overcome the inflation due to the sample relatedness, a kinship matrix was included as a random effect variable in the LMM. To capture latent confounders of gene expression levels, the first 5 PEERs were included in the regression model, where the PEERs were calculated using the peer (v1.3)[84] package under R.

For all eQTL analyses, SNPs that were most significantly associated with the gene expression were reported as raw p-values. Then, as done in previous studies[13,83], the *cis*-permutation pass of each SNP-gene pair was determined by correcting for the number of association tests, which overcomes the linkage disequilibrium problem by 1000 permutations. The significant eGenes were identified by performing q-value correction on the empirical *p*-values of the most significantly associated SNP-gene pair, using *q*-value < 0.05 as the threshold. The maximum *p*-value that passed the *q*-value threshold was used as the threshold to filter SNP-gene pairs and identify eVariants for an eGene. The above operations were done for each eQTL mapping run independently.

To test response eQTL, we incorporated three types of effects, namely, BCG effects, LPS effects, and LPS effects conditional on BCG effects, where the simulations were BCG, LPS, and LPS/BCG, respectively. In concrete, the BCG effects were evaluated by comparing post-vaccination (3 months) samples to pre-vaccination ones, where the samples were both treated by RPMI. LPS effects were assessed by comparing LPS-treated to RPMI-treated samples collected pre-vaccination. The LPS effects, conditional on BCG effects were tested on samples at 3 months post-vaccination, where the samples were treated by either LPS or RPMI. For each effect, a two-step method was used to define eGenes and eVariants. Firstly, in step 1, an interaction term between genotypes and conditions (GxC) was included in the LMM and evaluated for each type of effect. Genes significantly associated with the GxC term were reported as candidate eGenes, where the significance was defined by thresholds of corrected *q*-value < 0.05 or raw *p*-value < 5e-6. Then, in step 2, we estimated the effect size of genetic variants for each condition and identified eGenes using the same thresholds as used in step 1. Finally, loci were reported as significant response eQTL if (1) the SNP-gene pair was prioritized in step 1, and (2) the prioritized SNP was significantly associated with the expression of the paired gene upon stimulation but not at baseline. The prioritized SNP-gene pairs are available on Zenodo (https://doi.org/10.5281/zenodo.10949375), and full summary statistics are available online (https://lab-li.ciim-hannover.de).

## Kinship matrix construction

To account for the repeated measurement and donor relationships, we incorporated kinship as a co-variable in the eQTL mapping model. The high-quality and less redundant genetic variants obtained in the section "Genotyping, imputation, and quality control" were further pruned using PLINK (--maf 5e-2 --hwe 1e-6 --hwe-all --indep-pairwise 250 50 0.2) (version v1.90b6.26 64-bit (2 Apr 2022))[85]. Finally, we calculated a relationship matrix as the kinship matrix using PLINK tools with default parameters (--make-rel square).

## Identifying TI eQTL

TI was defined as the immune response differences between after and before being trained. Therefore, we first calculated the expression changes between T3m-LPS and T0-LPS based on each gene in each cell. Genes that are significantly differentially expressed in monocytes were defined as TI-response genes (FDR-adjusted-*p*-value < 0.05 and |log2(fold-change)| > 0.15), which resulted in 120 TI-response genes for the downstream analysis. In order to improve power and reduce inflation, we used the single-cell mode of limix_qtl with the same parameters described above. The single-cell mode uses the expression profile of each individual cell and includes a kinship matrix to release the burden of repeated measurements. To estimate the correlation between the identified TI-eVariant and TI-related pathways, the gene module score for each selected pathway was calculated using the ModuleScore() function from Seurat package[80], using default parameters. To release the concerns that the observed correlation between gene module scores and *LCP1*, we calculated gene module scores using 1000 random gene sets and calculated the correlation between the obtained scores and *LCP1* alterations (Supplementary Fig. 2e).

## Shared eQTL signals across cell types

The identified eQTL of five major cell types in four conditions enables us to estimate shared genetic effects under multiple treatments by the R implementation of multivariate adaptive shrinkage (MASH), MashR (version 0.2.57)[27], which uses the empirical Bayes hierarchical model. To obtain the robust, shared eQTL signals, we applied MASH using all eVariants identified in our consistent eQTL and response eQTL analyses. And meanwhile, we used the strong-random strategy to release the calculation costs. The strong model was constructed using the top eVariant of an eGene, and the data-driven covariance was estimated using five PCs. While the random model was built on 5000 randomly selected eVariants. Finally, the proportion of (significant) signals shared by magnitude in each pair of conditions was calculated using the factor of threshold 0.6.

## Colocalization analysis

We also estimated the potential shared causal variants between identified sc-eQTL and traits (e.g., common diseases and general traits) by a colocalization analysis. The R/coloc package (version 5.1.0.1)[68] was used to estimate the posterior probability of *H4* (PP4) that indicates the shared genetic effects between two traits from independent GWAS or QTL analyses. In concrete, we first harmonized datasets by matching the genetic variants from each of our eQTL analyzes to GWAS summary statistics downloaded from public domains, using harmonize_data() from the TwoSampleMR package (version 0.5.6)[86,87]. The publicly available GWAS summary statistics were collected from the IEU openGWAS database[88], which were listed in Supplementary Data 3. To speed up the processes, only SNPs that are associated with the gene expression at *p*-value < 0.5 were included for the harmonization. Next, the colocalization analysis was performed on every two paired sets of *p*-values from our sc-eQTL results and public GWAS using the coloc.abf() function with default parameters. Of note, we only analyzed loci containing SNP that were nominally associated with GWAS traits at a threshold of *p*-value < 5e-5. Finally, loci whose PP4 ≥ 0.5 were recognized as candidates of fair confidence and kept for downstream analysis (Supplementary Data 4).

### Independent genetic loci from public GWAS summary statistics

To estimate the shared genetic loci between our eQTL and publicly available GWAS loci, we performed a clump analysis to obtain independent GWAS loci using PLINK. The analysis exploited genotypes of Europeans from the 1000 Genome Project[89] as a reference panel using default parameters but specifying the following parameters, --clump-p1 5e-5 --clump-r2 0.1 --clump-kb 500. This resulted in a stringent list of independent loci for each GWAS summary statistics. The results are available in Supplementary Fig. 4a and Supplementary Data 5.

### SMR using summary statistics data

To evaluate if the effect of an SNP on complex traits is mediated by gene expression, we conducted an SMR (v1.3.1) analysis using gene expression as the exposure and GWAS traits as outcomes, where eVariants as instrumental variables, using default parameters[33]. The publicly available GWAS was obtained as described in the section "Colocalization Analysis". In the analysis, we conducted MR to identify potentially causal variants shared by exposure and outcome, meanwhile using the HEIDI (heterogeneity in dependent instruments) test to account for linkage that results in tangled association signals.

### Inhibiting gene expression experiments

Human peripheral blood mononuclear cells from healthy donors were isolated from buffy coats obtained from the Department of Transfusion Medicine at the University Hospital of Giessen and Marburg (UKGM) and deidentified prior to use. Monocytes were isolated using Miltenyi CD-14 beads and MACS purification and cultured in X-Vivo 15 medium (Lonza) at 37 °C in a humidified atmosphere with 5% CO2. To inhibit PU.1, cells were incubated with inhibitor DB2313 (MedChemExpress) or DMSO for 2 h and then stimulated with LPS (100 ng/ml) for 4 h. In a similar experiment, STAT1 was inhibited using Ruxolitinib (MedChemExpress). RNA was isolated using TRIzol (Ambion)/Chloroform extraction. The purified RNA was treated with DNase I (Roche) in the presence of a recombinant RNase inhibitor (NEB) and then extracted with phenol/chloroform/isoamyl alcohol (PCI) (Roth). Concentrations were determined using a Nanodrop 2000 instrument. Expression changes were measured in all experiments using a two-step qRT-PCR approach: reverse transcription (High-Capacity cDNA Reverse Transcription Kit; Thermo Fisher) and qPCR (Luna Universal qPCR Master Mix; NEB) were performed according to the manufacturer's instructions. Fold changes, normalized to U6 snRNA, were calculated using the 2-ΔΔCT method.

### Statistical analysis

All the statistical analyses were performed in R or Python programming languages. All in-house scripts to generate results in the current work are publicly available at GitHub (https://github.com/CiiM-Bioinformatics-group/300bcg_sceQTL). Heatmaps and upset maps were created using the ComplexHeatmap (v2.12.1) package[90]. Visualization of identified eQTL was performed using the Gviz (version 1.40.1)[91]. The allelic TF binding affinities were obtained from the ADASTRA database (v5.1.3)[38]. The co-expression analysis was performed using CS-CORE (v1.0.1), which estimates cell type specific co-expression by explicitly modeling sequencing depth variations and measurement errors in single-cell RNA sequencing data[92].

### Reporting summary

Further information on research design is available in the Nature Portfolio Reporting Summary linked to this article.

## Data availability

The scRNA-seq (FASTQ files) and genotype data (VCF files) used for sc-eQTL identification in this study are available at the European Genome-phenome Archive (EGA) database under accession code EGAS00001006990[17,18]. The scRNA-seq data (FASTQ files), scATAC-seq data (FASTQ files), and genotypes data (VCF files) of from COVID-19 patients used in this study are available at EGA under accession code EGAS00001006559 and EGAS00001006560 (https://ega-archive.org/datasets/ EGAS00001006560)[25]. All the above mentioned scRNA-seq, scATAC-seq and genotype data are available under restricted access to comply with the European Union General Data Protection Regulation for the protection of privacy-sensitive data. Access can be obtained by applications via the EGA data access portal under the Data access policy for the Helmholtz Center for Infection Research, Center for Individualized Medicine. The timeframe for response to requests is within two weeks. The bulk eQTLs are from the eQTLGen database[5] (https://www.eqtlgen.org). The eQTL datasets from purified specific immune cells (DICE database[26]) is available by https://dice-database.org. Gene-enhancer interactions are from the GeneHancer database[37] (https://www.genecards.org). The allelic TF binding information is from the ADASTRA database[38] (https://adastra.autosome.org). Information to access publicly available GWAS summary statistics datasets used in the current study are available and detailed in the Supplementary Data 3. The processed summary statistics of identified sc-eQTLs in this study are available on publicly scientific data repository platform, Zenodo (https://zenodo.org/doi/10.5281/zenodo.10949374)[93].

## Code availability

All in-house scripts to generate results in the current work are publicly available at GitHub (https://github.com/CiiM-Bioinformatics-group/300bcg_sceQTL).

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

## Acknowledgements

We thank the individuals of the 300BCG cohort for their participation in this study. This work was supported by an ERC Starting Grant (948207), a Radboud University Medical Center Hypatia Grant (2018), the Deutsche Forschungsgemeinschaft (DFG; German Research Foundation) under Germany's Excellence Strategy - EXC 2155 project number 390874280, COFONI (COVID19 Research Network of the State of Lower Saxony) with funding from the Ministry of Science and Culture of Lower Saxony, Germany (14-76403-184, 14-76403-184-3), and the Lower Saxony Center for AI and Causal Methods in Medicine (CAIMed) grant to Y.L. C.-J.X. was supported by Lower Saxony MWK Sprung grant (19777006). M.G.N. was supported by an ERC Advanced Grant (833247) and a Spinoza Grant of the Netherlands Organization for Scientific Research. This project was also supported by the European Union's Horizon 2020 research and innovation program under the Marie Sklodowska-Curie grant agreement No. 955321 to Y.L., M.G.N., and J.B. Z.Z. was supported by Singh-Chhatwal-Postdoctoral Fellowship Program. W.L. is supported by the China Scholarship Council (202006320050). The Genotype-Tissue Expression (GTEx) Project was supported by the Common Fund of the Office of the Director of the National Institutes of Health, and by NCI, NHGRI, NHLBI, NIDA, NIMH, and NINDS. The data used for the analyses described in this manuscript were obtained from: the GTEx Portal on 07/30/2023. We also thank the following people who recruited donors, collected samples, or assisted in analyzing data. Simone Moorlag played a crucial role in patient recruitment and sample collection for the 300BCG. Thomas Krausgruber managed the ATAC-seq sample processing, and Lukas Folkman was instrumental in data processing and analysis.

## Author contributions

Y.L. conceptualized, supervised, and financed the whole project. Z.Z. analyzed the data, created figures/tables, and drafted the manuscript, with W.L., Q.Z., J.B.-B., and MZ's assistance. Y.L., Z.Z., W.L., and C.-J.X.

discussed and interpreted all results. L.N.S. and M.A. performed experimental validations. R.H. and C.B. analyzed ATAC-seq data. L.A.B.J., C.-J.X., M.G.N., and M.J.B. supervised the whole project and advised the structure of the manuscript. All co-authors read through and revised the manuscript, figures, and supplementary tables/figures.

## Funding

## Competing interests
L.A.B.J. and M.G.N. are scientific founders of TTxD and Lemba. The remaining authors declare no competing interests.

## Additional information

[1]Department of Computational Biology for Individualised Infection Medicine, Centre for Individualised Infection Medicine (CiiM), a joint venture between the Helmholtz-Centre for Infection Research (HZI) and the Hannover Medical School (MHH), Hannover, Germany. [2]TWINCORE, Centre for Experimental and Clinical Infection Research, a joint venture between the Helmholtz-Centre for Infection Research (HZI) and the Hannover Medical School (MHH), Hannover, Germany. [3]Institute for Lung Research, Philipps University, Marburg, Germany. [4]Department of Internal Medicine and Radboud Center for Infectious Diseases, Radboud University Medical Center, Nijmegen, the Netherlands. [5]CeMM Research Center for Molecular Medicine of the Austrian Academy of Sciences, Vienna, Austria. [6]Department of Medical Genetics, Iuliu Haţieganu University of Medicine and Pharmacy, Cluj-Napoca, Romania. [7]Institute of Artificial Intelligence, Center for Medical Data Science, Medical University of Vienna, Vienna, Austria. [8]German Center for Lung Research (DZL), Giessen, Germany. [9]Department of Gastroenterology, Hepatology and Endocrinology, Hannover Medical School, Hannover, Germany. [10]Department for Genomics & Immunoregulation, Life and Medical Sciences Institute (LIMES), University of Bonn, Bonn, Germany. [11]Department of Genetics, University of Groningen, University Medical Center Groningen, Groningen, the Netherlands. [12]Division of Computational Genomics and Systems Genetics, German Cancer Research Center (DKFZ), 69120 Heidelberg, Germany. [13]European Molecular Biology Laboratory, Genome Biology Unit, 69117 Heidelberg, Germany. [14]Oncode Institute, Utrecht, The Netherlands. [15]Cluster of Excellence Resolving Infection Susceptibility (RESIST; EXC 2155), Hannover Medical School, Hannover, Germany. [16]Lower Saxony center for artificial intelligence and causal methods in medicine (CAIMed), Hannover, Germany.
✉e-mail: Yang.Li@helmholtz-hzi.de

