## [Transparent Peer Review file · Nature Communications]

Unveiling genetic signatures of immune response in immune-related diseases through single-cell eQTL analysis across diverse conditions

Corresponding Author: Professor Yang Li

Version 0:

Reviewer comments:

Reviewer #1

(Remarks to the Author)

This manuscript presents single-cell-based eQTL analysis of immune cells which covered baseline state and lipopolysaccharide (LPS) challenge either before or after Bacillus Calmette-Guerin (BCG) vaccination. The experiment is well designed, and I think this is the valuable single-cell eQTL resource, especially response eQTLs, trained immunity (TI) eQTLs, and TI cytokine QTLs (cQTLs). However, the authors mainly discussed only specific TI eQTLs and response eQTLs (eg. LCP1 and SLFN5). It would be better to describe the overall characteristics of response eQTLs and TI eQTLs, and compare them with conventional or baseline eQTLs. Below are major and minor comments with regards to this manuscript.

Major Comments

1. The main strength of this resource is the inclusion of TI eQTLs and cQTLs. However, the authors only focused on LCP1-rs2806897 eQTL effect in TI eQTL analysis. The authors should present all of the 120 TI-response SNP-gene pairs, and interpret the functions of TI-response genes. TI is defined as long-term memories in innate immune cells. Therefore, I suggest the authors perform TI eQTL analysis in each cell type and compare TI eQTL effects across cell types. Additionally, I propose that the authors comprehensively compare the differences between TI eQTLs and conventional cis-eQTLs in monocytes and the associations between TI eQTLs and cQTLs to provide more detailed insight into the genetic regulatory mechanisms of TI. Furthermore, I think it is also important to identify genes whose expression levels vary with TI in each cell type, which increase the resource value of this manuscript.

2. TI eQTL effect of rs2806897 on LCP1 and IL6R expression were presented in fold change of their expression (Fig. 2B and D). The eQTL effect of rs2806897 on LCP1 and IL6R expression per condition per cell type should also be described as in Fig. S7A. In addition, I am concerned that Fig. 2F may not be correct. Since IL-6 is ligand and IL6R is receptor, IL-6 should be upstream of IL6R. The authors illustrate the association between LCP1 and IL6R in Fig. 2F. Other than a just association in terms of expression level, are there any previous reports showing their association? I am concerned that this conclusion cannot be drawn only from the expression association. Furthermore, CD55 variant eQTL effect on IL6R may not necessarily be monocyte-specific because TI cQTL effect via IL-6, a ligand, could affect various cell types. I suggest that authors evaluate TI eQTL effect of rs2806897 on LCP1 and IL6R across cell types and compare them.

3. The previous eQTL studies mainly focused on eQTLs at baseline in unperturbed condition. The authors performed response eQTLs, which I think is one of the key points of this study. However, the summary and interpretation of response eQTLs is quite limited. For example, the number eGenes in B were smallest in conventional eQTLs (Fig. 3A), but those in B_Tm were largest in response eQTLs (Fig. S4B), which were not discussed at all. I think that the result of shared eQTL signals across cell types and conditions is very interesting (Fig. S4C), but there is no deep interpretation in this manuscript. To see the merit of including stimulated cells in eQTL analysis, the authors could comprehensively compare between response eQTLs and baseline eQTLs across cell types. Additionally, the power for identifying eQTLs greatly depends on gene expression level. I also suggest the authors investigate the association between response eQTLs and gene expression levels by perturbation.

4. The authors examined the allelic TF binding affinities of CD55 variant in monocytes using ADAstra database, and concluded that the three TFs (i.e., SPI1, STAT1, and EP300) are potential key regulators of CD55 by co-expression analysis

and in-vitro analysis. I think that only monocytes-specific expression of these TFs does not necessarily mean that they are potential key regulators of CD55 in monocytes (Fig. 4E, Line 280-282), and they are not strongly co-expressed with CD55 (Fig. 4F). In addition, I could not figure out why only SPI1 was selected in Fig. 4G. I suggest the authors build and score the gene regulatory network and assess the effect of potential TFs on CD55 expression using SCENIC or other methods.

5. One of the main motivations for creating eQTL catalog is to understand GWAS signals. The authors conducted colocalization analysis between immune traits/diseases GWASs and conventional cis-eQTLs or response eQTLs. However, only the number of colocalizations were shown (Fig. 3C and Fig. S4D), and only representative colocalizations were described (Fig. 3F, Fig. 4B, and Fig. 5B). The authors should compare the results of colocalization between cis-eQTLs and response eQTLs and show examples of colocalization only with response eQTLs, which can highlight the value of context-specific eQTLs. In addition, the threshold of colocalization was too low ($PP4 \geq 0.35$), which should be at least >0.5 . Although SLFN5 variant was presented as a representative example (Fig. 5B), PP4 value was low and I don't think that it is suitable to use this result in the main discussion.

6. In this manuscript, the authors generally used GWAS summary statistics that are not up-to-date for colocalization analysis. The authors should use the latest GWAS summary statistics.

7. I evaluate the resource value of this study, and think that all the summary statistics (i.e., conventional, response, TI and cytokine eQTLs) should be publicly available for open science. Furthermore, it is desirable to deposit raw data of individual single-cell data and genotype data.

Minor Comments

8: In this manuscript, most of cell types are based on coarse annotation (L1 level in Azimuth annotation). However, DC clusters were finely annotated (L2 level in Azimuth annotation), and this fine annotation did not lead to a meaningful result. It would be more consistent to integrate mDC and pDC into DC cluster.

9. In Figs. 5C,D, more quantitative comparison of SLFN5 expression levels should be done.

10. This study included healthy volunteers from Western European ancestry. I propose the authors provide the figures of genetic relationship between the samples in this cohort and 1000 Genomes Project.

11. In, Line283: Fig. S8A-C is wrong. I think it is Fig. S10A-C.

(Remarks on code availability)

Reviewer #2

(Remarks to the Author)

This study utilized single-cell RNA sequencing to analyze PBMCs under steady-state conditions, following treatment with lipopolysaccharide, and both before and after BCG vaccination, and scanned for eQTLs and their colocalizations with genetic risk loci for immune-related diseases. The authors then focused on several eQTLs associated with immune modulators and discussed their potential regulatory mechanisms and implications in diseases using public functional genomics data. However the study failed to adequately consider the diversity of immune cell subsets during in vivo and ex vivo challenges, as well as their roles in health and disease. Previous research has demonstrated that eQTLs often colocalize with disease loci in specific immune cell subsets, and the context-specific associations between eQTLs in native immune cells and their acute state upon stimulation are particularly significant. Consequently, estimating the genetic effects on gene expression in five major cell populations without considering cell sub-lineages in this study may result in missing a substantial fraction of causal variations relevant to immune responses. Furthermore, the study's limited sample size (only 38 donors) renders it underpowered for detecting eQTLs. With such a limited sample size, it's challenging to generalize findings or identify robust genetic features related to LPS and/or BCG vaccination. Additionally, the scRNA-seq and genotype data used in this study seem to have been previously published by the same group based on the absence of data availability information, as well as the same sample sizes and scRNA-seq libraries reported in both the current manuscript and the previously published paper (PMID: 37155329). Overall, the lack of novelty and validation prevents the generalization of its major conclusions. I have the following detailed concerns regarding the limited insights generated.

Major issues:

1. The authors prioritized several top eQTLs using different publicly available databases but did not adequately explore the shared characteristics of these eQTLs in relation to BCG vaccination and/or LPS treatment. Highlighting these features is essential to enhance the value of this dataset and analysis as a useful resource.

2. The authors observed an average of over 80% identified eQTLs across cell types with the eQTLGen database. While this alignment is expected, given that eQTLs curated by eQTLGen were derived from whole-blood and PBMC datasets, it remains unclear how many of these "consistent" eQTLs also overlap in terms of both statistical significance and allelic differences with other publicly available eQTL datasets derived from purified/sorted specific immune cells. Additionally, do these eQTLs colocalize between summary statistics from different traits?

3. The authors should justify their selection of specific eQTLs discussed in the main text. The rationale for connecting

COVID-19 pathogenesis to LPS/BCG responses, particularly regarding the SLFN5 eQTL, appears weak. Notably, the activation of human T cells by LPS is monocyte-dependent and requires direct cell-cell communication. Is there evidence of this SNP associating with SLFN5 in reported eQTL datasets derived from LPS-treated monocytes? Such an association could suggest prioritization of intercellular interactions acting in trans.

4. The authors should explicitly acknowledge that the data used in the current study were previously published. In schematic Figures 1A and 2A, authors should include references for published datasets and specify the number of individuals involved in each analysis.

5. Drawing conclusions for validation from results lacking statistical significance in lines 288-293 and 324-327 is confusing and raises concerns about the interpretation.

Additional suggestions and minor issues:

1. An potential intriguing avenue for investigation would be to explore cell-cell communications using scRNA-seq profiles and tools like CellPhoneDB. These analyses may help identify genetic determinants underlying immune responses relevant to LPS and/or BCG response.

2. Authors should provide justification for using a loose PP4 cutoff of 0.35 to indicate a likely causal association between eQTL and disease traits in the colocalization analysis.

3. Exploring whether the response eQTLs are primarily driven by variations in immune cell abundance would be interesting.

4. In line 172, it is unclear how the authors define the 120 TI-response genes. The Methods section lacks clear information on this, as well as on the 'transcriptome scores' mentioned in line 186.

5. In Fig. 2E, the bars represent p values, whereas the color indicates correlation strength—a discrepancy from the figure legend.

6. The correlations between LCP1 and the TI-relevant pathway genes in Figure 2F appear weak. The authors could assess whether these correlations occur by chance or are indeed enriched by comparing them with other pathways

7. Figure panels supporting the results in lines 334-335 of the main text are missing.

8. In line 388-340 (typo Fig.5D should be corrected to Fig.5F), the authors should clarify how they evaluated the significance of allelic imbalance and provide supporting evidence.

9. Typos in Fig.3E and S5A: "rs11687089_GT" should be corrected to "rs11687089_TC"; line 361 and Fig.2C: 'cQTL' should be 'eQTL'.

10. Figures S10 and S11 are not referred to in the main text.

(Remarks on code availability)

Version 1:

Reviewer comments:

Reviewer #1

(Remarks to the Author)

The authors have generally done a good job for addressing the reviewer's comments.

But I still have two remaining concerns about this study.

One of my concerns is how the eQTL sharing was assessed in Fig. S4E and S4F. While I acknowledge that assessing eQTL sharing between conditions is not straightforward, simply presenting heatmaps of eQTL effect sizes is not quantitative and persuasive. I would suggest formally modelling the overlapping/sharing, by using a method such as mash (DOI: 10.1038/s41588-018-0268-8), which will take into account both under which conditions the eQTLs are significant and whether the effect sizes are consistent. I believe a more quantitative evaluation will increase the value of this study.

The second one is colocalization analysis for at least COVID-19 and T2D GWAS. B2 (hospitalized COVID-19) and C2 (SARS-CoV-2 reported infection) phenotypes have been mainly highlighted in COVID-19 HGI GWAS consistently from release5 to the most recent release7 (DOI: 10.1038/s41586-023-06355-3) because they had more power to detect genetic signals than A2 (very severe COVID-19). From Table S3, I assume that the authors performed colocalization using A2 (of not latest one? Latest one was published in 2023.) in Fig 5B. Colocalization analysis using B2 or C2 of release7 might result in higher PP4. As for T2D, there is much larger GWAS (DOI: 10.1038/s41586-024-07019-6), which could lead to more robust

result in Fig S12G. Since these colocalizations are highlighted in this study, it would be desirable to use larger GWASs.

(Remarks on code availability)

Reviewer #2

(Remarks to the Author)

The authors have addressed my comments and made the necessary revisions. There is one minor issue: the missing legend for Fig.S2E.

(Remarks on code availability)

Version 2:

Reviewer comments:

Reviewer #1

(Remarks to the Author)

The authors have fully addressed my comments. Thank you.

(Remarks on code availability)

Reviewer #1 (Remarks to the Author)

This manuscript presents single-cell-based eQTL analysis of immune cells which covered baseline state
and lipopolysaccharide (LPS) challenge either before or after Bacillus Calmette-Guerin (BCG)
vaccination. The experiment is well designed, and I think this is the valuable single-cell eQTL resource,
especially response eQTLs, trained immunity (TI) eQTLs, and TI cytokine QTLs (cQTLs). However, the
authors mainly discussed only specific TI eQTLs and response eQTLs (eg. LCP1 and SLFN5). It would be
better to describe the overall characteristics of response eQTLs and TI eQTLs, and compare them with
conventional or baseline eQTLs. Below are major and minor comments with regards to this manuscript.

**A:** We sincerely thank the reviewer for their thoughtful comments on the overall document and we
agree with their suggestion to systematically describe response eQTLs and TI eQTLs. We have carefully
addressed all of the reviewer's comments as outlined below.

Major Comments

1. The main strength of this resource is the inclusion of TI eQTLs and cQTLs. However, the authors only
focused on LCP1-rs2806897 eQTL effect in TI eQTL analysis. The authors should present all of the 120
TI-response SNP-gene pairs, and interpret the functions of TI-response genes. TI is defined as long-
term memories in innate immune cells. Therefore, I suggest the authors perform TI eQTL analysis in
each cell type and compare TI eQTL effects across cell types. Additionally, I propose that the authors
comprehensively compare the differences between TI eQTLs and conventional cis-eQTLs in monocytes
and the associations between TI eQTLs and cQTLs to provide more detailed insight into the genetic
regulatory mechanisms of TI. Furthermore, I think it is also important to identify genes whose
expression levels vary with TI in each cell type, which increase the resource value of this manuscript.

We thank the reviewer for recognizing the main strengths of our study. Please find our point-by-point
responses to each of the comments below.

**Q:** The authors should present all of the 120 TI-response SNP-gene pairs, and interpret the functions
of TI-response genes.

**A:** We thank the reviewer's insightful comment. In response to the suggestion, we have included an
additional supplementary figure panel depicting the identified TI eQTL (now **Fig. S2A**, also on the next
page), and have reordered the previous panels to match the sequence in which they were referenced
in the manuscript. Furthermore, the summary statistics of the identified TI eQTL have been made
publicly available, facilitating further analysis and exploration by the broader scientific community (see
**Code and data availability** section).

Regarding "the biological functions of the identified TI-response genes", we provided a comprehensive
description and discussion in our previous work (PMID: **37155329**). In the current manuscript, our
focus is on TI eQTLs and their contribution to interindividual variation in TI response from a
transcriptional perspective. To clarify this focus, the following text has been added to the revised
manuscript on **lines 173 – 176, page 6**:

"To address this, we initially identified 120 TI-response genes and mapped TI eQTL by associating
genetic variants with expression alterations, cytokine production fold-change (FC) between T3m-LPS
and T0-LPS, of these TI-response genes (Fig. 2A and S2A, Methods)."

Fig. S2A The overview of eQTL effects of previously identified TI genes. The x-axis is the effect size
 and the y axis is the corresponding eGene. Top eVariant ($|\beta| \geq 0.3$) were labeled with the rsID
 and the example TI eQTL LCP1-rs2806897 were colored in red.

**Q:** *TI is defined as long-term memories in innate immune cells. Therefore, I suggest the authors perform*
*TI eQTL analysis in each cell type and compare TI eQTL effects across cell types.*

**A:** We appreciate the reviewer's suggestion to compare the eQTL across cell lineages, as this is a well-
established approach for illustrating genetic effects across transcriptional landscapes of different cell
types in typical single-cell or single-tissue eQTL studies.

As noted by the reviewer and described in the introduction of our manuscript, trained immunity (TI)
refers to the long-term memories in **innate immune cells**, leading to an enhanced immune response,
such as increased cytokine production, evoked by subsequent homologous or heterologous
restimulation. In our study, we focused on genes that show differential expression in monocytes
between T3m-LPS and T0-LPS, which we defined as TI-response genes. We then correlate the
expression changes of each of these genes with genotypes, defining these association as TI eQTLs.

We chose to focus on monocytes for TI eQTL analysis because the number of other innate immune cell
types in our dataset was generally low, limiting the statistical power. For example, neutrophils were
not well captured using the 10X Genomics' single cell RNA-seq technology. Additionally, the limited
number of NK cells present in the PBMC made it impractical to analyze them for TI effects. As a result,
we concentrated on monocytes both for identifying TI response gene in our initial study (PMID:
**37155329**) and for investigating TI eQTLs in the current study. This rationale is discussed in the
**Discussion** section on **lines 524 – 530** on **page 17** (also provided below).

"We recognize that exploring the TI effect in less abundant innate cell types, such as NK cells, will
require a larger sample size. Future studies could leverage single-cell analysis combined with cell
sorting or enrichment strategies to examine TI effects in neutrophils and other low-abundance innate
immune cells (PMID: **35618845**, PMID: **35545678**, and PMID: **35389779**)."

**Q:** *Additionally, I propose that the authors comprehensively compare the differences between TI eQTLs*
*and conventional cis-eQTLs in monocytes and the associations between TI eQTLs and cQTLs to provide*
*more detailed insight into the genetic regulatory mechanisms of TI.*

**A:** We thank the reviewer for the valuable suggestion and agree that the comparisons are indeed of
significant interest. In response, we have compared TI eQTLs to conventional/response cis-eQTLs in
monocytes. Of note, we observed intriguing patterns in effect size and direction between response
eQTL and TI eQTL.

The following text and figure have been added to the revised manuscript at **lines 258 – 264** on **page 9**:

"Finally, the effect sizes and directions of TI eQTLs show correlated patterns with those of response
eQTLs but show less similarity with the effect and directions of consistent eQTL in monocytes (Fig. S4F).
Given that TI assesses the differential response between LPS stimulation with BCG training (T3m-LPS)
and without BCG training (T0-LPS), it is noteworthy that genetic effect on TI (the difference between
T3m-LPS and T0-LPS) align with the genetic effect on the LPS response (either T3m-LPS or T0-LPS) for
these identified TI genes."

Fig. S4F Comparison of eQTL effects for the identified TI response genes. The x-axis represents the
 names of TI gene, while y-axis indicates the eQTL conditions. The colors indicate the effect sizes of the
 eQTLs.

Regarding “the comparison between TI eQTLs and cQTLs”, we conducted this analysis in the original
 manuscript. Our approach involved determining whether the significant TI eQTL were also associated
 with TI cQTLs. Among these shared loci, we identified *LCP1*-rs2806879 as an example to explore the
 role of TI eQTL in response to the secondary stimulation. We did not adopt a straightforward strategy
 of intersecting these the cQTL and TI eQTLs sets, as there were no genome-wide significant loci (with
 p-value cutoff of 5e-8) associated with cytokine production (cQTLs). This lack of association may be
 attributed to the limited sample size, which restricts the direct comparison of cQTLs with TI eQTLs.

To facilitate the exploration of our result, we have added a supplementary table (table S1) that includes
 the selected TI-cQTL and the rest tables were renamed accordingly. Additionally, to clarify our
 methodology, we have added the following text in the revised manuscript on line 178 on page 6:

“We compared the identified TI eQTL with TI cQTL effects obtained from the same cohort (table S1).”

**Q:** Furthermore, I think it is also important to identify genes whose expression levels vary with TI in
 each cell type, which increase the resource value of this manuscript.

**A:** We appreciate the reviewer for highlighting this important point, which we have addressed earlier
 in this response letter (please see page 3).

To summarize, we have systematically detailed the differentially expressed genes in response to TI in
 our previous work (PMID 37155329). In the current study, we focused on monocytes for evaluating TI
 eQTL effects due to their abundance in the dataset. The limited representation of other innate immune
 cells, such as NK cells or neutrophils constrained our ability to analyze them comprehensively. This
 limitation has been thoroughly discussed in the 'Discussion' section on lines 524 – 530 on page 17,
 and is also available in this response letter (lines 61 – 64 on page 3 of current document).

2. TI eQTL effect of rs2806897 on *LCP1* and *IL6R* expression were presented in fold change of their
 expression (Fig. 2B and D). The eQTL effect of rs2806897 on *LCP1* and *IL6R* expression per condition
 105 per cell type should also be described as in Fig. S7A. In addition, I am concerned that Fig. 2F may not
 be correct. Since *IL-6* is ligand and *IL6R* is receptor, *IL-6* should be upstream of *IL6R*. The authors
 illustrate the association between *LCP1* and *IL6R* in Fig. 2F. Other than a just association in terms of
 expression level, are there any previous reports showing their association? I am concerned that this
 conclusion cannot be drawn only from the expression association. Furthermore, *CD55* variant eQTL
 effect on *IL6R* may not necessarily be monocyte-specific because TI cQTL effect via *IL-6*, a ligand,

could affect various cell types. I suggest that authors evaluate TI eQTL effect of rs2806897 on LCP1
and IL6R across cell types and compare them.

Please find our point-by-point response below.

**Q:** The eQTL effect of rs2806897 on LCP1 and IL6R expression per condition per cell type should also
be described as in Fig. S7A.

**A:** We appreciate the reviewer's suggestion and thus have examined the eQTL effects of these two
genes using a similar strategy employed in Fig. S7A. In the R-Figure 1, neither LCP1 (panel A) nor IL6R
(panel B) showed significant trained immunity eQTLs effects in these cell type. Please see the figure
below for details.

**R-Figure 1.** Expression alterations of LCP1 (A) and IL6R (B) in CD4+ T, CD8+ T, NK cell and B cells. **A)**
Potential TI eQTL effects of LCP1-rs2806897 in other cell types. The x-axis is genotype groups and the
y-axis is the log2-fold-change of LCP1 gene expression. P-value and Z-score were included in the panel
title. **B)** Potential TI eQTL effects of IL6R-rs2806897. The x-axis is genotype groups and the y-axis is the
log2-fold-change of IL6R gene expression. P-value and Z-score were included in the panel title.

In response to the suggestion, we have added the following two figures into the manuscript (Fig. S2C
 and Fig.S2D).

Fig S2C. Boxplot showing expression levels of *LCP1* per genotype per cell type per condition.

Fig S2D. Boxplot showing expression of *IL6R* per genotype per cell type per condition.

**Q:** In addition, I am concerned that Fig. 2F may not be correct. Since IL-6 is ligand and IL6R is receptor,
IL-6 should be upstream of IL6R. The authors illustrate the association between LCP1 and IL6R in Fig.
2F. Other than a just association in terms of expression level, are there any previous reports showing
their association? I am concerned that this conclusion cannot be drawn only from the expression
association.

**A:** We thank the reviewer for highlighting this important point. We have revised Fig. 2F to better
illustrate the relationship between IL6 and IL6R. Specifically, we now show how IL6 binds to ILR, as
depicted below:

**Fig. 2F** Schematic plot demonstrating regulatory programs of *LCP1* in TI context.

Additionally, we have summarized literature evidence on potential interplays among IL6, IL6R and LCP1.
This summary is provided below and can also be found in the revised manuscript (lines 415 – 430,
page 14).

“Previous studies have suggested a potential interplay among IL-6, IL6R (IL-6 receptor) and LCP-1 in
various contexts [PMID: 32810692, PMID: 32318063, PMID: 35294888, PMID: 22904305]. IL-6 is well
known for its role as a warning signal in response to infections, tissue damages, or other stimuli [PMID:
25190079]. Upon binding to IL-6R, downstream signaling molecules such as gp130 activate the
JAK2/STAT3 pathway and JAK-SHP-2/MAPK pathway, which are central communication nodes for cell
functions. Recent studies on osteosarcoma involving the *LCP1* gene (also known as actin-bundling
protein L-plastin or LPL), have shown that the aberrant expression of *LCP1* gene activates the
JAK2/STAT3 signaling pathway, linking IL-6, IL-6R, and LCP1 in the context of tumor progression [PMID:
32810692]. Furthermore, LCP1 deficiency in mice is associated with decreased IL-6 production at six
150 hours post-infection with *P. aeruginosa* [PMID: 32318063]. This deficiency particularly affects the
151 myeloid compartment, making it more susceptible to infection. Moreover, LCP1 also plays a role in
modulating NLRP3-mediated production of IL-1 β and IL-18 by enhancing NLRP3 assembly in tissue-
resident macrophages [PMID: 35294888]. Since IL-6 is a known downstream target of IL-1 β , it is
consistently increased in serum from patients with NLRP3 inflammasome-mediated conditions [PMID:
22904305]. Collectively, these studies highlight potential interplays among LCP1, IL-6 and IL-6R, in line
with our observations.”

**Q:** Furthermore, CD55 variant eQTL effect on IL6R may not necessarily be monocyte-specific because
TI cQTL effect via IL-6, a ligand, could affect various cell types. I suggest that authors evaluate TI eQTL
effect of rs2806897 on LCP1 and IL6R across cell types and compare them.

**A:** Following the reviewer’s suggestion, we compared the TI eQTL effect of rs2806897 on *LCP1* and *IL6R*
across the remaining major cell types. As shown in R-Figure 1 (page 5 of this document), we did not
observe any significant association between gene expressions and genotypes in the other four major
cell types, including CD4+ T, CD8+ T, B cells, and NK cells. The differences were tested using linear mixed

model implemented in R/lme4 using the donor as the random effect variable, following the same
approach as in the original analysis.

3. The previous eQTL studies mainly focused on eQTLs at baseline in unperturbed condition. The
authors performed response eQTLs, which I think is one of the key points of this study. However, the
summary and interpretation of response eQTLs is quite limited. For example, the number eGenes in B
were smallest in conventional eQTLs (Fig. 3A), but those in B_Tm were largest in response eQTLs (Fig.
S4B), which were not discussed at all. I think that the result of shared eQTL signals across cell types
and conditions is very interesting (Fig. S4C), but there is no deep interpretation in this manuscript. To
see the merit of including stimulated cells in eQTL analysis, the authors could comprehensively
compare between response eQTLs and baseline eQTLs across cell types. Additionally, the power for
identifying eQTLs greatly depends on gene expression level. I also suggest the authors investigate the
association between response eQTLs and gene expression levels by perturbation.

We thank the reviewers' comments. Please find our point-by-point response below:

**Q:** *However, the summary and interpretation of response eQTLs is quite limited. For example, the*
*number eGenes in B were smallest in conventional eQTLs (Fig. 3A), but those in B_Tm were largest in*
*response eQTLs (Fig. S4B), which were not discussed at all. I think that the result of shared eQTL signals*
*across cell types and conditions is very interesting (Fig. S4C), but there is no deep interpretation in this*
*manuscript*

**A:** We sincerely thank the reviewer for his/her insightful feedback and for recognizing the importance
of our findings related to response eQTLs. We acknowledge the reviewer's observation regarding the
limited discussion of the number of eGenes in response eQTLs and the shared eQTL signals across cell
types and conditions. In response, we have expanded the discussion of these aspects to provide a
deeper interpretation, as summarized below.

First, we would like to emphasize that the primary focus of our study is on the interpretation of
regulatory programs and their disease relevance across three types of identified sc-eQTLs: consistent
eQTLs (e.g., ADCY3), opposing eQTL effects between cell types (e.g., CD55), and response eQTLs (e.g.,
SLFN5), which underscores values of identified eQTLs in various contexts. Through the integration of
these sc-eQTLs with multiple omics datasets and GWAS data, we provide novel insights into disease
mechanisms, building upon previous studies (PMID: 29610479, PMID: 35672358, PMID: 35618845),
which offer broader summaries of eQTLs, including shared and response eQTLs.

To clarify the reviewer's specific concerns:

**1. eGene distribution in response eQTLs in B cells:** We acknowledge that the number of eGenes in the
B cell subset is smallest in conventional eQTLs (Fig. 3A) but largest in response eQTLs (Fig. S4B). Note,
the original Fig. S4B has been relabeled to Fig. S4A. This difference is indeed significant and reflects
the dynamic response of B cells to stimuli. In line with established findings (PMID: 10482357, PMID:
18355243), LPS stimulation triggers B cell proliferation and differentiation into antibody-secreting cells,
which likely accounts for the observed expansion of eGenes in response eQTLs. We have now included
a more detailed discussion of this point in the revised manuscript at lines 242 – 246 on page 8.

"Interestingly, we acknowledge that the number of eGenes in the B cell subset is smallest in consistent
eQTLs (Fig. 3A) but largest in response eQTLs (Fig. S4A). This is in line with established findings (PMID:
10482357, PMID: 18355243), LPS stimulation triggers B cell proliferation and differentiation into

antibody-secreting cells, which likely accounts for the observed expansion of eGenes in response
eQTLs.”

**2. Shared eQTL signals across cell types and conditions:** We agree with the reviewer that the shared
eQTL signals across cell types (original **Fig. S4C**, current **Fig. S4B**) are particularly intriguing, as they
suggest a common genetic architecture underlying gene regulation. We have expanded on this
observation in the manuscript at **lines 251 – 256 on page 8 – 9.**

“Besides, we observed shared eQTL signals across cell types and conditions (**Fig. S4B**). The shared eQTL
signals across cell types and conditions suggest a conserved genetic architecture that governs gene
expression regulation. This implies a broader regulatory influence of these loci across diverse biological
contexts, highlighting their potential roles in coordinating molecular pathways and cellular functions
across different cell types and states.”

These insights add depth to our understanding of how genetic variants exert consistent effects across
related cell types, as well as how they might diverge under different conditions.

Moreover, following the reviewer’s suggestion, we have looked into the response eQTL result and
provided one additional example of response eQTL of *FBXL22* in B cells. These results have been
updated in the manuscript on **page 13 (lines 395 – 397, also available as the following).**

“Moreover, we also observed response eQTLs that are colocalized with complex traits involved in hosts’
immune responses, such as type 2 diabetes (**Fig. S12F and S12G**).”

**Fig. S12F.** Response eQTL effects of *FBXL22* gene. There are six panels in three columns. Each columns
represents the gene expression alteration upon stimulations from one condition. In the dot plots, the
x-axis are treatments, either “BL” or “Stim”, and the y-axis is the gene expression level. The dots
represent one donor and colors indicates genotypes. The box plots represent the gene expression per
genotype with x-axis and y-axis showing genotypes and gene expressions, respectively; the color
indicates Conditions which matches the to the top dot plots.

Fig. S12G. Colocalization of *FBLX22* eQTL and type II diabetes. The x-axis is the genome positions in
 base-pairs and the y-axis is the p-value transformed by $-\log_{10}()$. The colors of dots indicate eQTL (red)
 or GWAS (blue). The top SNPs from two summary statistic were labeled and the colocalization H4 and
 H3 were also labeled (top left corner).

In summary, we have expanded the manuscript to address the reviewer’s concerns and provide a more
 comprehensive interpretation of the response eQTLs and shared eQTL signals across cell types. Our
 study integrates sc-eQTL data with public epigenomic resources and multi-omics profiles, offering a
 detailed framework for understanding genetic regulations in disease contexts, particularly immune-
 related diseases.

**Q:** *To see the merit of including stimulated cells in eQTL analysis, the authors could comprehensively*
 *compare between response eQTLs and baseline eQTLs across cell types.*

**A:** We appreciate the reviewer’s suggestion to compare response eQTL with baseline eQTL, as this
 highlights the importance of including stimulated conditions in eQTL analysis. In response, we
 examined the shared loci between response eQTLs and consistent eQTLs across different cell types.
 Consistent with the systematic comparisons reported in previous studies (e.g. PMID: 35618845, PMID:
 35545678, and PMID: 35389779), we observed a similar pattern in our data (Fig. S4E). The results are
 summarized in newly added Fig S4E (shown below) and discussed on lines 248 – 251 on pages 8 – 9.

“The comparison between response eQTLs and consistent ones suggested specificity in the consistent
 eQTLs relative to response eQTLs (Fig. S4E). The response eQTLs identified under different stimulation
 conditions (LPS response eQTL before BCG or after BCG) exhibited notable similarity.”

**Fig. S4E.** Heatmap displaying the eQTL effects of identified sc-eQTLs. The x-axis represents the
 conditions for each cell type, while the y-axis lists eQTLs that were significant in at least one condition
 from one cell type. The colors indicate eQTL effects (beta coefficients). Additionally, example eQTLs are
 labelled along the y-axis.

**Q:** Additionally, the power for identifying eQTLs greatly depends on gene expression level. I also suggest
 the authors investigate the association between response eQTLs and gene expression levels by
 perturbation.

**A:** We appreciate the reviewer's insightful comment on the importance of gene expression levels in
 identifying eQTLs. Response eQTL effects are indeed determined and estimated based on gene
 expression changes, whether up-regulated or down-regulated, induced by stimulation. As such, we
 expect a natural correlation between response eQTLs and stimulation-induced gene expression
 changes.

As defined in our manuscript and previous studies (e.g. PMID: **29610479**), response eQTLs are
 characterized by SNPs associated specifically with baseline gene expression or gene expression upon

stimulation. This highlights that response eQTLs inherently reflect their relationship with gene
expression levels under perturbation.

In response, we looked into our data and observed 10.56% – 29.17% of differentially expressed genes
(adjusted p-value < 0.05) are response eGenes (FDR < 0.05) between the corresponding condition and
baseline. This information has been included in the revised manuscript at **lines 238 – 240 of page 8**,
which is also available the following.

“Interestingly, 10.56% – 29.17% of differentially expressed genes (adjusted p-value < 0.05) are
response eGenes (FDR < 0.05) between the corresponding condition and baseline (Fig. S4D).”

**Fig. S4D.** DE genes that were also identified as eGenes upon perturbations. The x-axis is the number
of genes (eGene or DE genes) while the y-axis is the conditions. The color of the bar indicates cell types
and the labels are percentage (exact number) of DE genes that were also identified as eGenes.

We hope this additional context and data address the reviewer’s suggestion.

4. The authors examined the allelic TF binding affinities of CD55 variant in monocytes using ADASTRA
database, and concluded that the three TFs (i.e., SPI1, STAT1, and EP300) are potential key regulators
of CD55 by co-expression analysis and in-vitro analysis. I think that only monocytes-specific expression
of these TFs does not necessarily mean that they are potential key regulators of CD55 in monocytes
(Fig. 4E, Line 280-282), and they are not strongly co-expressed with CD55 (Fig. 4F). In addition, I could
not figure out why only SPI1 was selected in Fig. 4G. I suggest the authors build and score the gene
regulatory network and assess the effect of potential TFs on CD55 expression using SCENIC or other
methods.

**Q:** I think that only monocytes-specific expression of these TFs does not necessarily mean that they are
potential key regulators of CD55 in monocytes (Fig. 4E, Line 280-282), and they are not strongly co-
expressed with CD55 (Fig. 4F).

**A:** We appreciated the reviewer’s insightful comments on prioritizing master TFs regulating CD55 and
agree that monocytes-specific expression does not necessarily imply monocyte-specific regulations of
CD55.

In response, we have performed additional prioritization of TFs using the gene regulatory network
analysis and revised the text at **lines 317 – 326 (page 11)** to describe the newly updated **Fig. 4F**, or

added **S9B** and **S9C**. Accordingly, the original co-expression figure (original **Fig. 4F**) has been moved to
the supplementary material (**Fig. S10A**). Details of this revision are provided in the subsequent point-
to-point response (**line 297 – 358** in this document).

**Q:** *In addition, I could not figure out why only SPI1 was selected in Fig. 4G. I suggest the authors build*
*and score the gene regulatory network and assess the effect of potential TFs on CD55 expression using*
*SCENIC or other methods.*

**A:** We thank reviewer’s insightful suggestion. In response, we have analyzed the regulon activities and
included these findings in the revised manuscript (**lines 317 – 326, page 11**), along with the updated
**Fig. S9B** and **S9C**; the original **Fig. S9** is now **Fig. S9A**. These analyses revealed cell-type-specific regulon
activities of SPI1 and its regulation of *CD55* under various conditions in monocytes. Further details are
provided in the manuscript as described below.

“Among the prioritized TFs, we assessed their effects on *CD55* expression using regulon analyses (PMID:
**28991892**). TF regulon enrichment, assessed via the AUC (Area Under the Receiver Operating
characteristic Curve), revealed that *SPI1* was specifically enriched in monocytes (**Fig. 4F**), while other
TFs did not display cell-type-specific over-representation (**Fig. S9B**). Furthermore, of the evaluated TF
and regulons, only *SPI1*, *STAT1*, and *ESR1* were predicted regulating *CD55*. And the *CD55* was among
the first of four quantiles of *SPI1*’s target genes which was ranked by the regulation potentials of *SPI1*
(**Fig. S9C**). Notably, a recent study (PMID: **39355884**) demonstrated that *SPI1* modulates *CD55*
promoter activities by binding to the rs2564978-C allele, consistent with our results.”

**Fig 4F.** Violin plot displaying SPI1 regulatory activities in multiple cells. The x-axis represents cell types
and the y-axis represents the AU cell score of SPI1 calculated by SCENIC (PMID: **28991892**).

Fig. S9B. UMAPs shows estimated AU cell score by SCENIC (PMID: 28991892) for the prioritized TFs.
 Each panel represents the AU cell score of a TF and the color indicates AU cell score.

Fig. S9C. The estimated importance of TF-targets in each TF regulon that is specifically bound to the
 rs2564978-C allele. Each panel indicates the regulatory capacity of a given TF in one condition. The x-
 axis indicates the estimated importance score by SCENIC (PMID: 28991892), while the y-axis indicates
 the rank of the regulatory target by the TF. *CD55* was labelled and its rank was included in the labels.

Finally, we also adjusted the schematic plot of Fig. 4G by including evidence from SCENIC analysis. (see
below).

Created in BioRender. Zhan, Q. (2025) <https://BioRender.com/c28r787>

* This study and Zhou et. al.
** Allele specific TF binding by Abramov et. al.
*** This study, Zhang et. al., and Schulte-Schrepping et. al.

**Fig. 4G** Schematic plot showing regulatory programs underlying *CD55*-rs2564978.

5. One of the main motivations for creating eQTL catalog is to understand GWAS signals. The authors
conducted colocalization analysis between immune traits/diseases GWASs and conventional cis-eQTLs
or response eQTLs. However, only the number of colocalizations were shown (Fig. 3C and Fig. S4D),
and only representative colocalizations were described (Fig. 3F, Fig. 4B, and Fig. 5B). The authors should
compare the results of colocalization between cis-eQTLs and response eQTLs and show examples of
colocalization only with response eQTLs, which can highlight the value of context-specific eQTLs. In
addition, the threshold of colocalization was too low ($PP4 \geq 0.35$), which should be at least >0.5 .
Although *SLFN5* variant was presented as a representative example (Fig. 5B), $PP4$ value was low and I
don't think that it is suitable to use this result in the main discussion.

Please find our point-by-point response below.

**Q:** The authors should compare the results of colocalization between cis-eQTLs and response eQTLs
and show examples of colocalization only with response eQTLs, which can highlight the value of
context-specific eQTLs.

**A:** For all the eQTL (including response eQTLs) discussed in the current manuscript, we focus on cis-
ones: SNPs that are in a window of 1M to the gene body were selected and evaluated, which had been
detailed in the Method section of the manuscript.

We agree that more examples of colocalization only with response eQTLs can highlight the value of
context-specific eQTLs, therefore, we have added another response eQTL example of *FBXL22* from B
cells (check page 9-10 of this documents). A full list of colocalization between response eQTLs and
diseases/traits were provided as **table S4** (page 23, line 705 of revised manuscript).

**Q:** In addition, the threshold of colocalization was too low ($PP4 \geq 0.35$), which should be at least >0.5 .
Although *SLFN5* variant was presented as a representative example (Fig. 5B), $PP4$ value was low and I
don't think that it is suitable to use this result in the main discussion.

**A:** We thank the reviewer for the insightful comments regarding the $PP4$ cutoff, which we also carefully
considered during our analysis. Following the reviewer's suggestion, we have updated our analyses to
use a more conventional $PP4$ cutoff of 0.5. This revision has been applied to previous **Fig. 3C** and **Fig.**
**S4D**. The original **Fig. S4D** has been relabeled to **Fig. S4B**. We also update the manuscript at **lines 703**
**– 704** on **page 23** to reflect the revised $PP4$ cutoff.

In the example of *SLFN5*, our original aim was to explore the regulatory program of *SLFN5* gene and
the colocalization result with $PP4 = 0.398$ was used as one line of evidence supporting its relevance in
COVID-19. This was complemented by literature on its role in viral infection (PMID: **36465279**, PMID:

33432153, PMID: 34556855). Importantly, our conclusion on *SLFN5* was derived from integration of
 multiple data layers, rather than the colocalization result based on solely PP4 value. The *SLFN5* serves
 as a robust example due to 1) its known involvement in viral infection, 2) replicated eQTL effects
 observed in a COVID-19 cohort, 3) alignment of its eQTL effect with allelic open chromatin status, and
 4) its association with COVID-19 at a suggestive p-value cutoff (p-value < 5e-5). Therefore, we retained
 the colocalization result in **Fig. 5B** as it serves as secondary evidence supporting the prioritization the
 *SLFN5* in this context.

We again thank the reviewer’s comments, and we willing to take further advice to improve this part.

6. In this manuscript, the authors generally used GWAS summary statistics that are not up-to-date for
 colocalization analysis. The authors should use the latest GWAS summary statistics.

**Q:** *The authors should use the latest GWAS summary statistics.*

**A:** We appreciate the reviewer’s suggestion to use the most up-to-date GWAS summary statistics for
 our colocalization analysis. Updating these resources is indeed critical for enhancing the robustness
 and relevance of our findings. During the preparation of this manuscript, we carefully selected GWAS
 summary statistics based on several key criteria, including their release date (restricted to no older
 than ten years), the number of loci, availability of full summary statistics, study cohort characteristics
 (adult Caucasian populations), and more.

To the best of our knowledge, the majority of the GWAS data we used were the latest available at the
 time of our analysis. For example, we used summary statistics from studies on Alzheimer’s diseases
 (2022), colorectal cancer (2020), and COVID-19 (2022). These release years have been added to **table**
 **S3** (table S2 before revision) in revised manuscript for reference (also below). However, we did include
 a few datasets older than ten years where newer, publicly available data were unavailable. For example,
 we included the 2013 GWAS summary statistics of obesities, as our analyses were conducted between
 2022-2023, and the criteria for selection were met by this dataset. Although a more recent GWAS on
 obesity was published in 2022 (PMID: 36187959), the full summary statistics were not publicly
 accessible, and the reported traits were obesity within the context of COVID-19. We hope this clarifies
 our approach and the reasoning behind our choice of GWAS summary statistics.

**Table S3** Years of GWAS summary statistics used in the current study

Trait	ID	PublicationYear
AlzheimerDiseases	ieu-b-5067	2022
AsthmaDT1	ukb-d-J10_ASTHMA	2018
AsthmaDT2	UKB-a-255	2017
AtopicDermatitis	ieu-a-996	2014
BladderCancer	ieu-b-4874	2021
BodyMassIndex	ieu-b-40	2018
COVID19Release4	ebi-a-GCST010780	2020
COVID19Release7	COVID19_HGI_A2_ALL_eur_leave23andme	2022
COVID19Release7	COVID19_HGI_A2_ALL_leave_23andme	2022
ColorectalCancer	ieu-b-4965	2021
CoronaryHeartDisease	ieu-a-7	2015
CrohnsDisease	ieu-a-30	2015

DiastolicBloodPressure	ieu-b-39	2018
GoutDisease	UKB-a-107	2017
HDLCholesterol	ieu-b-109	2020
Height	ieu-a-89	2014
InflammatoryBowelDisease	ieu-a-31	2015
LDLCholesterol	ieu-b-110	2020
LungCancer	ieu-a-966	2014
MultipleSclerosis	UKB-b-17670	2018
ObesityClass1	ieu-a-90	2013
ObesityClass2	ieu-a-91	2013
ObesityClass3	ieu-a-92	2013
OvarianCancer	ieu-b-4963	2021
ProstateCancer	ieu-b-4809	2021
Psoriasis	UKB-a-100	2017
RheumatoidArthritis	ukb-d-M13_RHEUMA	2018
Schizophrenia	ieu-b-42	2014
SystemicLupusErythematosus	EBI-a-GCST003156	2015
SystolicBloodPressure	ieu-b-38	2018
ThyroidCancer	ieu-a-1082	2013
Triglycerides	ieu-b-111	2020
Type2Diabetes	EBI-a-GCST006867	2018
UlcerativeColitis	ieu-a-32	2015
Urate	met-a-345	2014

7. I evaluate the resource value of this study, and think that all the summary statistics (i.e., conventional,
response, TI and cytokine eQTLs) should be publicly available for open science. Furthermore, it is
desirable to deposit raw data of individual single-cell data and genotype data.

**A:** We appreciate the reviewer’s emphasis on open science and fully agree with the importance of
making research data publicly accessible. but are sorry for missing the publicly available deposition
information in the manuscript. The summary statistics are available via Zenodo
(<https://zenodo.org/doi/10.5281/zenodo.10949374>) and the private information, including genotypes
and raw sequencing data, are available via reasonable requests by accession number
EGAS00001006990 on European Genomic Archives (EGA). Theses information has been also included
in the **Data availability and Code availability** section of the updated manuscript at **lines 754 – 766** on
**page 25** (check below).

“Single-cell RNA-seq data and genotype data have been deposited at European Genome-phenome
Archive (EGA), which is hosted by the EBI and the CRG, under accession number EGAS00001006990
(<https://ega-archive.org/studies/EGAS00001006990>). GWAS summary statistics used in the analyses
are detailed in table S3.”

“All in-house scripts to generate results in current work are publicly available at GitHub
(https://github.com/CiiM-Bioinformatics-group/300bcg_sceQTL). Summary statistics of identified
eQTLs are available on publicly scientific data repository platform Zenodo

<https://zenodo.org/doi/10.5281/zenodo.10949374>). Any additional information required to
reanalyze the data reported in this paper is available from the lead contact upon request at any time
available.”

Minor Comments

8: In this manuscript, most of cell types are based on coarse annotation (L1 level in Azimuth
annotation). However, DC clusters were finely annotated (L2 level in Azimuth annotation), and this fine
annotation did not lead to a meaningful result. It would be more consistent to integrate mDC and pDC
into DC cluster.

**A:** The suggestion to merging mDC and pDC into DC totally makes sense to us. The figures involving
DCs has been updated by merging pDC and mDC into DC cells. However, we made a comprise by
removing DC cells in our analysis, considering the relatively low proportions of DC cells (both pDC and
mDC). The cell-type annotations are trustable: the reference-based annotations were further manually
checked by visualizing the expression of known marker genes. And we updated the figures and
manuscripts following the reviewer’s suggestion to merge pDC and mDC into DCs. The updated figures
include **Fig 1B/1C/1E, Fig S1E/S1F/S1G**, while the updated text includes **lines 156 – 158 on page 5,**
**line 516 on page 17, and lines 595 – 596 on page 20.**

9. In Figs. 5C,D, more quantitative comparison of SLFN5 expression levels should be done.

**A:** We are sorry for the lack of statistical notations on the **Fig 5C and 5D**, which were exclusively
described in the manuscript text. Therefore, we update the figures by adding hypothesis test p-values
from Seurat package. And the following text were added the figure legend for the clarification purpose.

“The asterisks or dots indicate statistical significances: ‘.’ P > 0.05, ‘***’ P < 0.001, ‘*****’ P < 0.0001.”

10. This study included healthy volunteers from Western European ancestry. I propose the authors
provide the figures of genetic relationship between the samples in this cohort and 1000 Genomes
Project.

**A:** We highly appreciate this suggestion considering our cohort is a group of Caucasian individuals. The
PCA plot indicating genetic relationship across samples has been included as a supplementary figure
(**Fig. S1C**), while the panels in **Fig S1** and legends were reordered and the figure labels were also
adjusted in the updated manuscript. Finally, we add the following paragraph to depict the genetic
relationship across samples, which is available at **lines 145 – 147 on page 5.**

“Of note, the genotype PCA plots including donors of current study and 1000 genome (phase 3) donors
suggested a fair homogeneity of the study cohort (Fig. S1C).”

Fig. S1C PCA plot displays the genetic relationship between the current study cohort and 1000 genome
cohort. The colors of the markers suggest sub-populations and the shape of the markers indicate the
source of the donors.

11. In, Line283: Fig. S8A-C is wrong. I think it is Fig. S10A-C.

**A:** We apologize for miscounting and mis-labeling the figures. These have been fixed in the updated
manuscript at line 328 on page 11. The figures have been relabeled to Fig. S10 due to the modifications
mentioned in the current documents.

Reviewer #2 (Remarks to the Author)

This study utilized single-cell RNA sequencing to analyze PBMCs under steady-state conditions,
following treatment with lipopolysaccharide, and both before and after BCG vaccination, and scanned
for eQTLs and their colocalizations with genetic risk loci for immune-related diseases. The authors then
focused on several eQTLs associated with immune modulators and discussed their potential regulatory
mechanisms and implications in diseases using public functional genomics data. However the study
failed to adequately consider the diversity of immune cell subsets during *in vivo* and *ex vivo* challenges,
as well as their roles in health and disease. Previous research has demonstrated that eQTLs often
colocalize with disease loci in specific immune cell subsets, and the context-specific associations
between eQTLs in native immune cells and their acute state upon stimulation are particularly
significant. Consequently, estimating the genetic effects on gene expression in five major cell
populations without considering cell sub-lineages in this study may result in missing a substantial
fraction of causal variations relevant to immune responses. Furthermore, the study's limited sample
size (only 38 donors) renders it underpowered for detecting eQTLs. With such a limited sample size,
it's challenging to generalize findings or identify robust genetic features related to LPS and/or BCG
vaccination. Additionally, the scRNA-seq and genotype data used in this study seem to have been
previously published by the same group based on the absence of data availability information, as well
as the same sample sizes and scRNA-seq libraries reported in both the current manuscript and the
previously published paper (PMID: 37155329). Overall, the lack of novelty and validation prevents the
generalization of its major conclusions. I have the following detailed concerns regarding the limited
insights generated.

Please find our point-by-point response below.

**Q:** *However, the study failed to adequately consider the diversity of immune cell subsets during *in vivo**
*and *ex vivo* challenges, as well as their roles in health and disease. Previous research has demonstrated*
*that eQTLs often colocalize with disease loci in specific immune cell subsets, and the context-specific*
*associations between eQTLs in native immune cells and their acute state upon stimulation are*
*particularly significant. Consequently, estimating the genetic effects on gene expression in five major*
*cell populations without considering cell sub-lineages in this study may result in missing a substantial*
*fraction of causal variations relevant to immune responses.*

**A:** We sincerely thank the reviewer for the insightful comments. We fully agree that considering the
diversity of immune cell subsets it is crucial for elucidating genetic effects and their associations with
disease loci. The specific point regarding the colocalization of eQTLs with disease loci in specialized
immune cell subsets and context-specific associations is specifically well-taken.

The primary focus of our study was to provide a comprehensive overview of genetic effects on gene
expression across major immune cell populations. This foundational analysis is an important step
toward understanding broad genetic influences under both *in-vivo* and *in-vitro* stimulation conditions,
contributing to a better understanding biological processes such as trained immunity or disease
mechanisms before exploring sub-lineage-specific variations in details.

In our study, we demonstrated the value of **integrating of sc-eQTL results with** other data – such as
trained immunity cytokine QTL, GWAS summary statistics, ATAC-seq data of the same cohort, allele-
specific transcription factor binding potentials, and scRNA-seq/scATAC-seq profiles of diseases (e.g.
COVID-19) - can yield novel insights and enhance our understanding of genetic regulation across these

major cell types. Additionally, recent studies have made significant contributions by analyzing eQTLs
in specific immune cell subpopulations (e.g. PMID: 35618845, PMID: 35545678, and PMID: 35389779).
These works provide valuable resources and detailed perspectives that complement our broader
approach, highlighting that comprehensive sub-lineage analyses are indeed being actively pursued.

Given the current literature and the scope of our study, our approach aimed to balance depth and
breadth, creating a platform that can be extended in future investigations to apply our **integration**
**methodology** to more specific cell subpopulations under varied conditions

To address this important aspect, we have included the following text in the Discussion section of the
revised manuscript (**lines 525 – 530 on page 17**):

“Future studies could leverage single-cell analysis combined with cell sorting or enrichment strategies
to examine TI effects in neutrophils and other low-abundance innate immune cells [PMID: 35618845,
PMID: 35545678, and PMID: 35389779]. Applying the integration approach demonstrated in this study
to such high-resolution datasets would enable researchers to gain a more comprehensive
understanding of context-specific eQTLs and their associations with disease.”

**Q:** *Furthermore, the study’s limited sample size (only 38 donors) renders it underpowered for detecting*
*eQTLs. With such a limited sample size, it’s challenging to generalize findings or identify robust genetic*
*features related to LPS and/or BCG vaccination.*

**A:** We agree that the sample size of this study is relatively small compared the above three example
studies. Therefore, we used a few strategies to release its impact:

- 1) stringent criteria to filter variants, in which we ensured each genotype group have at least
three donors;
- 2) a linear mixed model was used to integrate multiple samples from the same donor while
remove the random effects, for examples, expanding the sample size into 152 for consistent
eQTL and 76 for response-eQTL analyses, respectively;
- 3) we validated our eQTL in publicly available eQTL databases to ensure our eQTLs are trustable;
- 4) when interpreting the examples, we introduced multiple published datasets to complement
our findings.

We think these operations help to release the concerns on the limited number of samples. They also
help to improve the possibility of generalizing our findings and robustness of the identified genetic loci.

**Q:** *Additionally, the scRNA-seq and genotype data used in this study seem to have been previously*
*published by the same group based on the absence of data availability information, as well as the same*
*sample sizes and scRNA-seq libraries reported in both the current manuscript and the previously*
*published paper (PMID: 37155329). Overall, the lack of novelty and validation prevents the*
*generalization of its major conclusions.*

**A:** We sincerely thank the reviewer’s comments and would like to take opportunity to clarify our work.

In response to the comments regarding to data availability, we apologize for any confusion in the initial
submission. We have addressed this by adding more detailed information in the revised manuscript
(**page 26, lines 778 – 786**), which is also available at **lines 392 – 401 on page 17** of this document.

To clarify, while the scRNA-seq data for two conditions (T0_LPS and T3m_LPS) were previously analyzed
to describe the transcriptional response related to trained immunity effect response (PMID: 37155329),

this current study extends beyond that. Our analysis incorporates scRNA-seq data from **four distinct**
**conditions** (RPMI before vaccination, LPS before vaccination, RPMI after vaccination, LPS after
vaccination), encompassing both *in vivo* (BCG vaccination) and *in vitro* (LPS) stimulations. Furthermore,
we integrated genome-wide genotype data from the same donors (only in this study but not in the
previous one PMID: **37155329**). This manuscript specifically focuses on the genetic regulation of
transcriptional changes within the context of trained immunity and under various stimulation
conditions.

We hope this clarification addresses the reviewer's concerns and highlights the novel aspects of our
study. Additionally, we would like to emphasize the following novel aspects of our work:

- • To our knowledge, this is the first study to evaluate the genetic impacts on gene expression at
the single cell level within the context of trained-immunity (TI). We identified TI eQTL, linked
these to TI cytokine productions and investigated the underlining regulatory mechanisms. This
approach enhances our understanding of TI at the transcriptional level. Additionally, it is also
the first report of sc-eQTLs using data from this cohort.
- • To demonstrate the value of this dataset for exploring regulatory mechanisms, we presented
examples of TI eQTLs, consistent eQTLs, and response-eQTLs as prototypes for dissecting
regulatory programs. Our study provides a unique resource by integrating eQTLs with multi-
modal data from the same donors, as well as publicly available datasets and patient data,
thereby showcasing the regulatory potential underlying eQTLs.

We appreciate the reviewer's feedback regarding the importance of validation and sincerely apologize
for not clearly conveying our extensive validation efforts in the original manuscript. We agree
wholeheartedly that validation is crucial, and we have employed multiple approaches to ensure the
robustness of our findings. These efforts included the use of independent datasets (either from
different measurement modalities or publicly available databases) as well as experimental validation.

We also would like to reiterate our interpretation of sc-eQTLs from the manuscript.

- 1. **LCP1 eQTL in TI context:** The *LCP1* eQTL example were firstly introduced considering TI is of of
our main focus of this manuscript. The TI eQTL has been further investigated using trained
immunity cytokine QTL data and its impact in TI context were supported by multiple statistical
analyses.
- 2. **ADCY3 eQTL in monocyte** (consistent eQTL): As shown in **Fig. 2**, *ADCY3* is an example of
consistent sc-eQTLs with functional relevance across conditions. Through Mendelian
Randomization (MR) and colocalization (Coloc) analyses, we integrated multiple GWAS traits
to consistently pinpointed the role of *ADCY3* in monocytes and its involvement in disease
mechanisms, demonstrating the utility of sc-eQTLs in uncovering cell-type-specific regulatory
mechanisms in diseases.
- 3. **CD55 eQTL across cell types with opposing effects.** Another key finding is the *CD55* eQTL,
which displayed an interesting pattern of opposite eQTL effect between monocytes and B
cells. Given *CD55*'s established role in the influenza context (PMID: **22693232**, PMID:
**28510725**, PMID: **34197564**), we sought to dissect the regulatory mechanism by integrating
data with allelic TF binding database (ADAstra) and estimating genetic effects on gene
expression. By employing allele-specific methods such as allelic TF binding analysis, we

uncovered the potential regulatory programs driving these opposite effects between cell
types. As shown in Fig. S7, the observed *CD55* eQTL effect is also present in T3m.
4. ***SLFN5* response eQTL.** We prioritized the *SLFN5* response eQTL through analysis of publicly
available datasets (including ATAC-seq, allelic TF binding database, and COVID19 GWAS data),
and further dissected its underlying regulatory program in lymphocytes using sc-eQTL from
healthy donors. We also used single-cell ATAC-seq (Assay for Transposase-Accessible
Chromatin with Sequencing) from COVID-19 patients to assess allele-specific open chromatin
accessibility. Additionally, previous studies (PMID: **36465279**) have established the role of
*SLFN5* in influenza infection, further supporting its involvement in infectious diseases. Our
analyses expanded potential candidate genes for COVID-19 and provided insights in the
epigenetic regulation of *SLFN5* in infection and immune responses. Importantly, the regulatory
effect of the eQTL SNP has been experimentally validated in the literature [PMID: **24604203**].

Of these examples, they were either replicated in independent studies or supported by multiple layers
of data. Therefore, we are confident in their future replications by independent research groups and
generalizations in other application context.

In summary, we hope these points effectively address the reviewer's concerns while highlight both
the innovative aspects and the rigorous validation of our study.

Major issues:

1. The authors prioritized several top eQTLs using different publicly available databases but did not
adequately explore the shared characteristics of these eQTLs in relation to BCG vaccination and/or LPS
treatment. Highlighting these features is essential to enhance the value of this dataset and analysis as
a useful resource.

**A:** We fully agree with the review's idea to dive into shared characteristics between the identified
eQTLs and BCG vaccination/LPS treatment. We have attempted to explore the potential commonality
using the following methods:

- 1. Direct comparison by intersecting these eQTLs across conditions (RPMI, BCG and LPS) and cell
types by **Fig. 3A** and **Fig. S4A** (in revised figures)
- 2. Shared eQTL signals by **Fig. S3D** and **S4B** in revised figures, estimated by MASH (PMID:
**30478440**)
- 3. Estimating whether they were colocalized with the same traits **Fig. 3C** and **S4C**.

These results suggested shared features of the identified eQTLs, such as specificity to cell-types in
response contexts. Moreover, the analyses to indicate the effects of BCG or LPS stimulations have been
added on **page 11** of this response letter (**Fig. S4E**) and the results have also been updated in the new
version of manuscript at **lines 237 – 256** on **page 8**. Secondly, in the initial manuscript we explored the
**shared genetic effect** on gene expression by performing **consistent eQTL** analyses where T0-RPMI, T0-
LPS, T3m-RPMI, and T3m-LPS conditions were combined using linear mixed model by limix
(<https://github.com/limix/limix>). On the other hand, the condition specific eQTLs were identified by
using interaction model with stimulation as interaction term. Details can be found in the revised
manuscript on **page 20 – 23**.

2. The authors observed an average of over 80% identified eQTLs across cell types with the eQTLGen
database. While this alignment is expected, given that eQTLs curated by eQTLGen were derived from

whole-blood and PBMC datasets, it remains unclear how many of these "consistent" eQTLs also
overlap in terms of both statistical significance and allelic differences with other publicly available eQTL
datasets derived from purified/sorted specific immune cells. Additionally, do these eQTLs colocalize
between summary statistics from different traits?

We fully agree with the reviewer's comments on the replication results and shared colocalization
signals across GWAS summary statistics from different traits. Please find our point-by-point response
below.

**Q:** *How many of these "consistent" eQTLs also overlap in terms of both statistical significance and allelic*
*differences with other publicly available eQTL datasets derived from purified/sorted specific immune*
*cells.*

**A:** We thank the reviewer's comments on the validation using eQTLGen database and agree that it is
important to validate the consistent eQTLs in datasets from purified/sorted immune cells. Following
the reviewer's suggestion, we have validated our findings using an independent eQTL databases, DICE
database, which is known as a comprehensive resource of purified immune cells ([https://dice-](https://dice-database.org)
database.org). These results were added to manuscript at **lines 222 – 224** on **pages 7 – 8** together with
**Fig. S3C.**

**Fig. S3C** Bar plot showing the number replicated eGenes in DICE database. The x-axis is the number of
eGenes while the y-axis are cell types. The color of each bar indicates replication in DICE database and
the number are exact number (percentage) of the bar per cell type.

“We also validated our eQTL findings in eQTL datasets from purified specific immune cells (DICE
database, PMID: 30449622), which resulted in more than 90% validation rates (Fig. S3C).”

**Q:** *Additionally, do these eQTLs colocalize between summary statistics from different traits?*

**A:** Following the reviewer's suggestion (a similar suggestion from reviewer 1), we also checked if the
identified eQTL are colocalized with known GWAS signals together. Please see our response on lines
207 – 232 on page 9 – 10 of this document.

3. The authors should justify their selection of specific eQTLs discussed in the main text. The rationale
for connecting COVID-19 pathogenesis to LPS/BCG responses, particularly regarding the SLFN5 eQTL,
appears weak. Notably, the activation of human T cells by LPS is monocyte-dependent and requires
direct cell-cell communication. Is there evidence of this SNP associating with SLFN5 in reported eQTL
datasets derived from LPS-treated monocytes? Such an association could suggest prioritization of
intercellular interactions acting in trans.

The authors appreciate the reviewer's suggestion. Please find our point-by-point response below.

**Q:** *The authors should justify their selection of specific eQTLs discussed in the main text. The rationale*
*for connecting COVID-19 pathogenesis to LPS/BCG responses, particularly regarding the SLFN5 eQTL,*
*appears weak.*

**A:** We apologize for the ambiguity in the description of evidences used to prioritize the example eQTLs
reported in the manuscript. The four eQTL examples described in the main text were carefully selected
to represent distinct categories: one TI eQTL, two consistent/conventional eQTLs (one monocyte-
specific and the other showing opposite eQTL effects across cell types), and one response eQTL. These
examples were chosen to cover all categories of eQTLs identified in this study.

Regarding the stimulation experiment, LPS (lipopolysaccharide) is commonly used as an *ex-vivo*
stimulant to mimic infection conditions, as it triggers an immune response similar to that induced by
infections. By stimulating cells with LPS in eQTL studies, we can observe gene expression changes
under simulated infection conditions, providing insights into how genetic variants may influence
immune function (PMID: **28814792**, PMID: **29610479**). In addition, with both BCG vaccination and LPS
stimulation, it allows us for the first time to evaluate the genetic effect on in-vivo trained immunity
induced gene expression changes at single cell level.

Importantly, our primary focus was on genetic regulation of *SLFN5*, which was based on the following
evidence:

- 1. validation of the eQTL in publicly available eQTL datasets, such as eQTLGen;
- 2. conventional methods to deconvolute GWAS signals, such as COLOC and Mendelian
Randomization;
- 3. chromatin open regions that from the same cohort, i.e., ATAC-seq results;
- 4. regulatory potentials from publicly available database such as GenHancer and ADASTRa.
Specifically, in the *SLFN5* example, we observed response eQTL which located on the promoter
region of the gene.

The connection of *SLFN5* to virus infection was based on a few publications (PMID: **36465279**, PMID:
**33432153**, PMID: **34556855**). The regulatory effect of SNP on *SLFN5* gene was experimentally validated
by point mutation using CRISPR/Cas systems (PMID: **24604203**). Moreover, the connection between
*SLFN5* and COVID19 was based on the following evidence:

**1)** Previous study reported the suggestive association between *SLFN5* locus and COVID-19 severity in
the updated COVID-19 GWAS (PMID: **35241825**). **2)** We replicated the eQTL effect of *SLFN5*-
rs11080327 in one independent COVID-19 cohort (PMID: **36474914**). **3)** In the COVID-19 cohort, we
observed the allelic imbalance of chromatin accessibility at the identified SNP, i.e., rs11080327 (PMID:
**36474914**).

Of note, the above observations are in line with each other and collectively revealed the role of *SLFN5*
in COVID-19, which were not systematically reported in previous studies. To improve the manuscript,
we have revised texts to reflect the above information and have adjusted the tone accordingly at **lines**
**470 – 477** on **pages 15 – 16** of the revised manuscript:

“Previous studies suggested that *SLFN5* is potentially involved in COVID-19 (PMID: **35241825**), however,
the underlying regulatory program is still unclear. Therefore, we uncovered the epigenetic and genetic
regulation mechanisms by integrating multiple evidences, including 1) colocalization by COLOC (PMID:

24830394) and Mendelian Randomization by SMR (PMID: 27019110); 2) open chromatin peak
 harboring the eVariant of *SLFN5* measured in the same individuals, i.e., ATAC-seq results; 3) regulatory
 potentials of identified eVariant from publicly available database such as GenHancer and ADAstra
 (PMID: 33980847).”

**Q:** Notably, the activation of human T cells by LPS is monocyte-dependent and requires direct cell-cell
 communication. Is there evidence of this SNP associating with *SLFN5* in reported eQTL datasets derived
 from LPS-treated monocytes? Such an association could suggest prioritization of intercellular
 interactions acting in trans.

**A:** We fully agree that it is interesting to estimate intercellular interactions acting in trans since the T-
 cell activation by LPS is partially monocyte-dependent. In concrete, previous study revealed that the
 LPS-induced T-cell activation is assisted by primed monocytes (PMID: 9531301), which is highly
 depended on the cell-cell communications. Additionally, T lymphocytes can also be activated by LPS
 via TLR4/CD14 complex (PMID: 19405031, PMID: 19166381, PMID: 17579019), which enhances the
 ability of T cells to beat bacteria.

Following reviewer’s suggestion, we have investigated the eQTL effect of rs11080327 on *SLFN5* in
 **monocytes** upon stimulations. We observed eQTL effects in monocytes in our dataset (**R-Figure 2**). Of
 note, this finding was validated in an independent eQTL dataset from purified classical monocytes in
 the DICE database which were recommended by the reviewer for replication (**R-Figure 3**).

**R-Figure 2.** Response eQTL effects of rs11080327-*SLFN5* in monocytes.

**R-Figure 3.** Response eQTL effects of rs11080327-*SLFN5* in monocytes from DICE database.

These results suggest potential intercellular interactions acting in trans between monocyte and T cells
in response to LPS stimulation. The precise direction of these *interactions* would require experimental
validation in future research. We have incorporated this intriguing observation and supporting data
into the revised manuscript (**lines 493- 501, pages 16 – 17**):

“Previous studies have shown that the LPS-induced T-cell activation is assisted by primed monocytes
(PMID: 9531301), which is highly depended on the cell-cell communications. Additionally, T
lymphocytes can also be activated by LPS via TLR4/CD14 complex (PMID: 19405031, PMID: 19166381,
PMID: 17579019), which enhances the ability of T cells to beat bacteria. To explore whether the eQTL
effect of rs11080327 on *SLFN5* is also present in monocytes under stimulation, we analyzed our dataset
and observed this eQTL effect in monocytes. These findings suggest potential intercellular interactions
acting in trans between monocyte and T cells in response to LPS stimulation. However, the precise
direction and mechanism of these interactions will require experimental validation in future studies.”

4. The authors should explicitly acknowledge that the data used in the current study were previously
published. In schematic Figures 1A and 2A, authors should include references for published datasets
and specify the number of individuals involved in each analysis.

**A:** We sincerely apologize for missing the information indicating some datasets used in the analyses
had been already published. The publication information was added to the updated manuscript in the
**Data availability and Code availability** sections of the updated manuscript at **lines 754 – 766 on page**
**25** (check below). And the figure legend has been also updated based on the reviewer’s comments
(**lines 789 – 790, page 27**). The updated information is also available as the following:

“Single-cell RNA-seq data and genotype data have been deposited at European Genome-phenome
Archive (EGA), which is hosted by the EBI and the CRG, under accession number EGAS00001006990
(<https://ega-archive.org/studies/EGAS00001006990>). GWAS summary statistics used in the analyses
are detailed in table S3.”

“All in-house scripts to generate results in current work are publicly available at GitHub
(https://github.com/CiiM-Bioinformatics-group/300bcg_sceQTL). Summary statistics of identified
eQTLs are available on publicly scientific data repository platform Zenodo
(<https://zenodo.org/doi/10.5281/zenodo.10949374>). Any additional information required to
reanalyze the data reported in this paper is available from the lead contact upon request at any time
available.”

5. Drawing conclusions for validation from results lacking statistical significance in lines 288-293 and
324-327 is confusing and raises concerns about the interpretation.

**A:** We totally agree with the reviewer’s comments that without statistical significance, no solid
conclusions can be drawn. We apologize for any confusions caused.

We acknowledge that our experimental evidence (**lines 334 – 336 in revised manuscript**) is suggestive,
likely due to limited sample size. However, we consider it as supplementary evidence. To validate the
regulation of SPI1 on *CD55*, we have gathered two independent lines of evidence.

1. The eQTL effects were validated in independent datasets including eQTLGen and DICE, both
at an adjusted p-value threshold of 0.05.

2. The allelic TF binding affinity was estimated in ADASTRA (PMID: **33980847**), and regulatory
interactions were confirmed using the GenHancer database.

While our experimental validation showed a trend of effect in the expected direction, it did not reach
statistical significance. Encouragingly, our observations are further supported by a recently published
paper (PMID: **39355884**). Specifically, Uvarova et al demonstrated that knocking down *SPI1* led to
balanced promoter activity of *CD55* between samples carrying minor alleles (T) and major allele (C) of
rs2564978, which is in line with our results. This information has not been incorporated in the revised
manuscript (**lines 334 – 336 on page 11**) as the following:

“Of note, a recent publication (PMID: **39355884**) revealed that rs2564978-C allele is associated with
increased promoter activities upon bound by SPI1, aligning with our results.”

For the result at lines 324-327 (line 373 in revised manuscript), we acknowledge that the suggestive
eQTL (rs11080327-*SLFN5*) p-value of 0.09 from validation analysis may cause some confusion, as it
does not reach the conventional significance threshold of 0.05. However, we would like to clarify that
we validated the rs11080327-*SLFN5* eQTL using *two* datasets:

- 1. Single-cell RNA seq from up to 50 COVID-19 patients (MHH50 cohort, PMID: **36474914**): This
dataset yielded a significant p-value of 4.55e-05. The results were further contextualized in
COVID-19 using multiple supporting datasets, including allelic TF binding affinities and allelic
open chromatin data.
- 2. Single-nuclei RNA-seq from 43 COVID-19/LC donors (*unpublished* data): In this dataset, the p-
value was 0.09, which we included as supplementary evidence to the first COVID-19 dataset.

The lack of significance in the second dataset could be attributed to key differences between the two
cohorts. For example:

- 1. Sequencing methods: The first COVID-19 cohort were profiled using single-cell RNA-seq while
the second one employed single-nuclei RNA-seq.
- 2. Sample size: the second dataset was relatively small, limiting its power to detect genetic effects
on gene expression (n = 43).

Building on the replication results from the first COVID-19 dataset, we further investigated allelic open
chromatin and allelic TF binding affinities to confirm the SNP’s effect in the disease context. Importantly,
the eQTL effect (rs11080327-*SLFN5*) and the causal effect of rs11080327-A allele on *SLFN5* are
consistent with previously reported findings (PMID: **24604203**, PMID: **36465279**).

Therefore, our conclusions were based on the collective evidences from multiple layers of data
(detailed information are available on **pages 20 – 23** of this response letter). Finally, to improve the
clarity, in the revised manuscript, we have added the following text at **lines 372 – 374 on page 12**:

“To sum up, the rs11080327-*SLFN5* eQTL effect was suggestively replicated in patient data across two
independent COVID-19 datasets, with P-value of 4.55e-05 and 0.09, respectively.”

Additional suggestions and minor issues:

- 1. An potential intriguing avenue for investigation would be to explore cell-cell communications using
scRNA-seq profiles and tools like CellPhoneDB. These analyses may help identify genetic determinants
underlying immune responses relevant to LPS and/or BCG response.

**A:** Indeed, the communications between cell lineages are crucial for the immune responses. As
mentioned in the major issue number 4, the data used in the current study was already published in
previous studies. Specifically, the cell-cell communications (CCC) analyses have been done in our
previous study using the same sing-cell sequencing datasets (PMID: 37155329).

2. Authors should provide justification for using a loose PP4 cutoff of 0.35 to indicate a likely causal
association between eQTL and disease traits in the colocalization analysis.

**A:** We agree with the reviewer's critical comments on the PP4 cutoff and the authors of the COLOC
package recommends using $PP4 \geq 0.5$ as a thumb of rule to determine if the association is causal.
Therefore, we updated our figures and main text by using the conventional cutoff of $PP4 \geq 0.5$. More
information is available on lines 407 – 414 on page 18 of the *current* response document.

3. Exploring whether the response eQTLs are primarily driven by variations in immune cell abundance
would be interesting.

**A:** We totally agree with the reviewer's idea to prove whether the response eQTL are primarily driven
by variations of immune cell abundance. This has been discovered and discussed in previous study
(PMID: 35672358), which is in line with our results. To clarify this point, we have added the following
text to underscore it at lines 240 – 242 on page 8 of the revised manuscript.

“We also observed correlations between number of identified responses eQTLs and abundance of a
cell type, which is in line with previous study (PMID: 35672358).”

4. In line 172, it is unclear how the authors define the 120 TI-response genes. The Methods section
lacks clear information on this, as well as on the ‘transcriptome scores’ mentioned in line 186.

**A:** We are sorry for the unclear definition of the TI-response genes. Thus, we added the following text
to the **Method** section to clarify the prioritization parameters at lines 664 – 667 on page 22 of revised
manuscript.

“Genes that are significantly differentially expressed in monocytes were defined as TI-response genes
(FDR adjusted p-value < 0.05 and \$|\log_2(\text{fold-change})| > 0.15\$ ), which resulted in 120 TI-response genes
for the downstream analysis.”

For the *transcriptome scores*, we have adjusted the term into “gene module scores” to clarify that it
ensembled expression value of a set of genes belong to the same pathway. And the following text were
added to the **Method** section at lines 670 – 673 on page 22 of the updated manuscript.

“To estimate the correlation between the identified TI-eVariant and TI-related pathways, gene module
score for each selected pathway was calculated using the ModuleScore() function from Seurat package,
using default parameters.”

5. In Fig. 2E, the bars represent p values, whereas the color indicates correlation strength—a
discrepancy from the figure legend.

**A:** We are sorry for the mistake which has been fixed in the updated manuscript.

6. The correlations between LCP1 and the TI-relevant pathway genes in Figure 2F appear weak. The
authors could assess whether these correlations occur by chance or are indeed enriched by comparing
them with other pathways.

**A:** Thanks for the reviewer’s critical comments on the correlation results. The correlation analyses are
 supplementary evidence where the *LCP1* was already defined as a TI-response gene. Following your
 suggestion, we evaluated whether the correlation between *LCP1* and selected pathways are random
 or not by adding a correlation analysis between *LCP1* expression alteration and module scores
 calculated using 1,000 random selected gene sets. We did not observe strong correlations between
 *LCP1* expression alteration and the module score from random selected genes (**Fig S2E**), which we
 believe is a descent proof that *LCP1* is indeed correlated to TI. The methods to generate the figure
 panel was also added to the updated manuscript at **lines 673 – 675** on **page 22** which is as the following:
 “To release the concerns that the observed correlation between gene module scores and *LCP1*, we
 calculated gene module scores using 1,000 random gene sets and calculated the correlation between
 the obtained scores and *LCP1* alterations (**Fig. S2E**).”

**Fig. S2E.** Permutation results from correlations between gene module scores from random selected
 gene sets and *LCP1* expression. The x-axis is the Spearman’s Rho and the y-axis is the frequency, where
 1,000 random gene sets were selected. The dashed blue is the 2.5% and 97.5% quantile of the
 Spearman’s Rho values from the permutation analysis. The rest-colored vertical lines indicated the
 Spearman’s Rho between gene module scores of TI-related pathways and *LCP1* expression.

7. Figure panels supporting the results in lines 334-335 of the main text are missing.

**A:** We are sorry for not making it clear and thanks for pointing it out. The detailed description (**lines**
 **1,131 – 1,139** on **pages 42 – 43** of revised manuscript) about the panel is available in figure legend for
 **Fig. 5E**, which is the following:

“Open chromatin regions of the identified eVariant associated with *SLFN5* expression in the COVID-19
 context. The top two tracks display the chromatin accessibility coverage in different CD4+ T cells
 partitioned by COVID-19 conditions (i.e., convalescent and hospitalized), where the coverage was
 normalized by the read counts in TSS. The “Peaks” track shows the open chromatin peaks which were
 connected to the expression of *SLFN5* in the “Peak2GeneLinks” track, where the gene structure is
 visualized in the bottom track. Moreover, the color of the connecting line suggests the correlation at a
 threshold of adjusted p-value < 0.05.”

8. In line 388-340 (typo Fig.5D should be corrected to Fig.5F), the authors should clarify how they
 evaluated the significance of allelic imbalance and provide supporting evidence.

**A:** We are sorry for the misleading reference to the figure panels. However, by this sentence, we meant
the allelic imbalance was in line with the eQTL effects and referred back to the **Fig. 5D** (the differential
expression analysis). The method to evaluate the allelic imbalance are available at Zhang et al (PMID:
**36474914**).

9. Typos in Fig.3E and S5A: “rs11687089_GT” should be corrected to “rs11687089_TC”; line 361 and
Fig.2C: ‘cQTL’ should be ‘eQTL’.

**A:** The typos were fixed about which we are sorry. However, for the one at line 361 as well as in Fig.2C,
they were indeed ‘cQTL’ instead of ‘eQTL’.

10. Figures S10 and S11 are not referred to in the main text.

**A:** We have referred these figures in the updated manuscript, for S10 at **line 328 on page 11**, and for
**S11 at lines 347 – 348 on page 12**.

Reviewer #1 (Remarks to the Author):

*The authors have generally done a good job for addressing the reviewer's comments.*

*But I still have two remaining concerns about this study.*

**Q:** *One of my concerns is how the eQTL sharing was assessed in Fig. S4E and S4F. While I acknowledge*
*that assessing eQTL sharing between conditions is not straightforward, simply presenting heatmaps of*
*eQTL effect sizes is not quantitative and persuasive. I would suggest formally modelling the*
*overlapping/sharing, by using a method such as mash (DOI: 10.1038/s41588-018-0268-8), which will*
*take into account both under which conditions the eQTLs are significant and whether the effect sizes*
*are consistent. I believe a more quantitative evaluation will increase the value of this study.*

**A:** We fully agree that a overall heatmap is not efficient enough to show the shared signals of eQTLs
across conditions. We thanks the reviewer's suggestion to quantify the shared eQTL signals using
MASH. Indeed, we applied MASH to model the overlapping/sharing and the results were presented
in both original and previously revised manuscript (lines 227 – 229 on page 8 in revised manuscript,
Fig. S3D and S3E).

"Additionally, a further multivariate adaptive shrinkage (MASH) analysis revealed that consistent eQTLs
from similar cell lineages exhibit comparatively greater similarity in the genetic effects on gene
expression (Fig. S3D and S3E)."

Moreover, the corresponding method are available at lines 677 – 687 on page 22 of the revised
manuscript of first revision.

"The identified eQTL of five major cell types in four conditions enables us to estimate shared genetic
effects under multiple treatments by the R implementation of multivariate adaptive shrinkage (MASH),
MashR (version 0.2.57), which uses the empirical Bayes hierarchical model. To obtain the robust,
shared eQTL signals, we applied MASH using all eVariants identified in our consistent eQTL and
response eQTL analyses. And meanwhile, we used the strong-random strategy to release the
calculation costs. The strong model was constructed using the top eVariant of an eGene and the data-
driven covariance was estimated using five PCs. While the random model was built on 5000
parameters. To release the concerns that the observed correlation between gene module randomly
selected eVariants. Finally, the proportion of (significant) signals shared by magnitude in each pair of
conditions was calculated using the factor of threshold 0.6."

**Q:** *The second one is colocalization analysis for at least COVID-19 and T2D GWAS. B2 (hospitalized*
*COVID-19) and C2 (SARS-CoV-2 reported infection) phenotypes have been mainly highlighted in COVID-*
*19 HGI GWAS consistently from release5 to the most recent release7 (DOI: 10.1038/s41586-023-06355-*
*3) because they had more power to detect genetic signals than A2 (very severe COVID-19). From Table*
*S3, I assume that the authors performed colocalization using A2 (of not latest one? Latest one was*
*published in 2023.) in Fig 5B. Colocalization analysis using B2 or C2 of release7 might result in higher*
*PP4. As for T2D, there is much larger GWAS (DOI: 10.1038/s41586-024-07019-6), which could lead to*
*more robust result in Fig S12G. Since these colocalizations are highlighted in this study, it would be*
*desirable to use larger GWASs.*

**A:** We totally agree that the updated and comprehensive COVID-19 and T2D GWAS summary statistics
will deepen our insights. Therefore, we have updated our manuscripts and figures by incorporating the
above up-to-date summary statistics following the reviewers suggestion.

For the COVID-19 analyses, we indeed initially used the A2 results (*HGI-A2-ALL-eur-leave23andme-*
*20220403*, release 7) of HGI COVID-19 GWAS summary statistics which have been officially available
on April 8, 2022 (<https://www.covid19hg.org/results/r7>). Following the reviewer’s suggestion, we have
performed colocalization analyses for B2 (*HGI_B2_ALL_eur-leave23andme-*20220403) and C2
(*HGI_C2_ALL_eur-leave23andme-*20220403) results.

For the type 2 diabetes, we have switched to the updated summary statistics by DIAGRAM study which
is the recommended one by the reviewer. More information are available at [https://www.diagram-](https://www.diagram-consortium.org/downloads.html)
[consortium.org/downloads.html](https://www.diagram-consortium.org/downloads.html).

And correspondingly, the texts at **line 396 on page 13** (highlighted in green), **Fig. 3C, S3F and S4C** (the
corresponding figure legend have also been updated and highlighted in green), and **supplementary**
**table 3** have been updated.

Next, we have compared the colocalization results between *SLFN5* eQTL and COVID19 GWAS results
(now including A2, **B2 and C2**). However, we *did not* observed colocalization signals for the B2 and C2
results, likely due to the absence of significant GWAS signals at this locus (R-Figure 1). This discrepancy
may stem from differences in case – control definitions used in GWAS analysis. Specifically, A2
represents “*Very severe respiratory confirmed COVID vs. population*”; B2 identifies genetic loci
associated with “*Hospitalized COVID vs. not hospitalized COVID*”, and C2 highlights potential genomic
hot-spots by comparing “*COVID vs. population*”. Given the lack of colocalization in B2 and C2, we
retained only the results based on A2 summary statistics in the main results.

**R-Figure 1.** Colocalization results between *SLFN5* eQTL (CD4T) and COVID-19 GWAS results.

Reviewer #2 (Remarks to the Author):

**Q:** *The authors have addressed my comments and made the necessary revisions. There is one minor*
*issue: the missing legend for Fig.S2E.*

**A:** We seriously apologize for this mistake. We have now added the following text to describe the figure,
which is also available at the next page of Fig. S2 in the revised manuscript.

“(E) Histogram plot showing the permutation test results for pathway scores.”